

**Considering the future of anthropogenic gas-phase organic compound**
**emissions and the increasing influence of non-combustion sources on urban air**
**quality**
Peeyush Khare [1] and Drew R. Gentner [1, 2, *]
[1] Department of Chemical and Environmental Engineering, Yale University, New Haven CT-06511 USA
[2] School of Forestry and Environmental Studies, Yale University, New Haven CT-06511 USA
* To whom correspondence may be addressed. Email: drew.gentner@yale.edu
Keywords: Anthropogenic emissions, volatile and semivolatile organic compounds (VOCs,
SVOCs), ozone, secondary organic aerosol (SOA), urban air quality, consumer products,
building materials
**Abstract**
Decades of policy in developed regions has successfully reduced total anthropogenic emissions of
gas-phase organic compounds, especially volatile organic compounds (VOCs), with an intentional,
sustained focus on motor vehicles and other combustion-related sources. We examine potential
secondary organic aerosol (SOA) and ozone formation in our case study megacity (Los Angeles),
and demonstrate that non-combustion-related sources now contribute a major fraction of SOA and
ozone precursors. Thus, they warrant greater attention beyond indoor environments to resolve
large uncertainties in their emissions, oxidation chemistry, and outdoor air quality impacts in cities
worldwide. We constrain the magnitude and chemical composition of emissions via several
bottom-up approaches using: chemical analyses of products, emissions inventory assessments,



theoretical calculations of emission timescales, and a survey of consumer product material safety
datasheets. We demonstrate that the chemical composition of emissions from consumer products,
and commercial/industrial products, processes, and materials is diverse across and within
product/material-types with a wide range of SOA and ozone formation potentials that rivals other
prominent sources, such as motor vehicles. With emission timescales from minutes to years,
emission rates and source profiles need to be included, updated, and/or validated in emissions
inventories, with expected regional/national variability. In particular, intermediate-volatility and
semivolatile organic compounds (IVOCs and SVOCs) are key precursors to SOA but are excluded
or poorly represented in emissions inventories, and exempt from emissions targets. We present an
expanded framework for classifying VOC, IVOC, and SVOC emissions from this diverse array of
sources that emphasizes a lifecycle approach over longer timescales and three emission pathways
that extend beyond the short-term evaporation of VOCs: (1) solvent evaporation, (2) solute off-
gassing, and (3) volatilization of degradation by-products. Furthermore, we find that ambient SOA
formed from these non-combustion-related emissions could be misattributed to fossil fuel
combustion due to the isotopic signature of their petroleum-based feedstocks.
**1. Introduction**
Anthropogenic emissions of gas-phase organic compounds, including volatile organic compounds
(VOCs), are of direct concern as toxic or carcinogenic air pollutants in indoor and outdoor
environments (Cohen et al., 2005; Nazaroff and Weschler, 2004; Weschler and Nazaroff, 2008).
Often they are more important for air quality as reactive precursors to the formation of outdoor
tropospheric ozone and secondary organic aerosol (SOA) as well as indoor SOA, and thus play a
central role in the elevated mortality and morbidity rates caused by fine mode particulate matter



(i.e. $PM_{2.5}$) and ozone in both developed and developing regions (Destaillats et al., 2006; Jerrett et
al., 2009; Lim et al., 2012; Nazaroff and Weschler, 2004; Pope and Dockery, 2006; Sarwar et al.,
2004; Singer et al., 2006; Weschler, 2011). In urban and downwind areas globally, 20-70% of
$PM_{2.5}$ is organic aerosol (OA), with SOA comprising 58% of OA in urban areas and 82%
downwind on average (Zhang et al., 2007).
Yields of SOA and ozone are strongly dependent on precursor molecular size, volatility, structure,
and moieties/functionalities as well as environmental conditions (Gentner et al., 2012; Kroll and
Seinfeld, 2008). SOA models have struggled to reproduce observations due to incomplete
knowledge on SOA precursors and their sources (Hallquist et al., 2009; Kroll and Seinfeld, 2008).
Gas-phase organic compound measurements and emissions data have historically focused on
VOCs (i.e. $C_2$-$C_{12}$ alkanes and VOCs with equivalent volatilities), but research has demonstrated
the prevalence and importance of larger, intermediate-volatility and semivolatile organic
compounds (IVOCs and SVOCs, respectively) (Gentner et al., 2012; Kroll and Seinfeld, 2008;
Robinson et al., 2007; Zhao et al., 2014). With greater SOA mass yields, IVOCs and SVOCs are
key for modeling and mitigating SOA but are less-studied due to measurement difficulties
(Gentner et al., 2012; Goldstein and Galbally, 2007). While their emissions from motor vehicles
have received more attention, emissions from most other sources are poorly constrained.
In recent history, motor vehicles, power plants, and residential combustion have been dominant
drivers of detrimental air quality (e.g., VOCs, nitrogen oxides ($NO_X$), $PM_{2.5}$, ozone, sulfur dioxide
($SO_2$)), with automobiles dominating anthropogenic VOC emissions in major cities (Gentner et



al., 2017; Hao et al., 2007; McDonald et al., 2013, 2015; Warneke et al., 2012; Zhou et al., 2003).
Yet due to the success of combustion-related emissions control policies over the past 60 years,
motor vehicle VOC (and other pollutant) emission factors have decreased by orders of magnitude
in route to today's diesel and gasoline vehicles, albeit with some recent on-road diesel $NO_X$
compliance issues (Hao et al., 2007; Kirchstetter et al., 1999; U.S. Environmental Protection
Agency, 2011; Warneke et al., 2012). In no other place has this been more studied than Los
Angeles, CA, where $PM_{2.5}$ and ozone still exceed standards, and OA comprises 41% of $PM_1$, 66%
of which is SOA (Hayes et al., 2013; Warneke et al., 2012). Recent work has suggested that motor
vehicle emissions in L.A. cannot fully explain observations of reactive precursors and SOA
(Ensberg et al., 2014; Hayes et al., 2015; McDonald et al., 2015; Zhao et al., 2014). In L.A. and
beyond, the role of motor vehicles in degrading air quality will decline further with the newest on-
road emissions standards (e.g. Tier 3/LEV-III, Euro 6/VI) (Figure 1), use of electric-powered
vehicles, and stricter regulations on emissions from off-road vehicles and other engines (Gentner
et al., 2017; Giannouli et al., 2011; Gordon et al., 2013; Platt et al., 2014; Tessum et al., 2014;
Weiss et al., 2011).

The main objective of this paper is to evaluate anthropogenic sources of gas-phase organic
compounds that are gaining prominence as motor vehicles and other combustion-related sources
become cleaner in the developed and developing world. We demonstrate that consumer products
and commercial/industrial products, processes, and materials are important, widely-distributed
area sources of VOCs, IVOCs, and SVOCs with variable source profiles making their emissions
and impacts difficult to observe and constrain. Our demonstration builds on relevant indoor air
quality studies observing emissions and substantial SOA formation from ozone-initiated reactions





with emissions from consumer products, building materials, and cleaning products (Chang et al.,
2011; Destaillats et al., 2006; Gold et al., 1978; Lewis, 2001; Mitro et al., 2016; Nazaroff and
Weschler, 2004; Singer et al., 2006; Weschler, 2011; Weschler and Nazaroff, 2008; Wilke et al.,
2004) as well as secondary emissions from oxidative degeneration of indoor paints, varnishes, and
materials (Knudsen et al., 1999; Poppendieck et al., 2007a, 2007b; Salthammer and Fuhrmann,

95    2007).


We outline a holistic framework to assist future research in the field, and then use a multi-dataset
approach to constrain emissions, which includes the most detailed emissions inventory available,
laboratory analysis of product composition, a survey of consumer product material safety
datasheets (MSDSs), and calculations of emissions timescales from theory. Specifically, we (a)
evaluate the pathways, chemical composition, and magnitude of VOC, IVOC, and SVOC
emissions from prominent types of products, processes, and materials; (b) calculate their potential
to form SOA and ozone; and (c) compare their emissions and potential air quality impacts to other
prominent sources. We identify major knowledge gaps in emission rates and composition,
including I/SVOCs, which we explore via two examples: pesticides and asphalt-related emissions.

**2. Defining a comprehensive framework for non-combustion-related emissions**
There is a wide body of work on some aspects of emissions from products, processes, and materials
in indoor or outdoor environments, but outdoor-focused emissions inventories are mostly focused
on evaporative emissions of VOC solvents at the time of application. Here we outline the current



scope of emissions and expand on it to define a comprehensive framework to guide future studies
on this class of emissions.

*Volatility:* Emissions of gas-phase organic compounds can and should be differentiated by their
volatility: VOCs, IVOCs, and SVOCs, or further by their n-alkane-equivalent volatility (Murphy
et al., 2014). One can broadly refer to compounds emitted from products and processes via the
prefix 'pp-' (e.g. pp-VOCs), which is consistent with existing conventions for volatility basis set
and aerosol abbreviations (e.g. pp-SOA) (Murphy et al., 2014).

*Source categories:* While the spectrum of sources is very diverse, they are often grouped by use.
The following source categories are consistent with the most detailed VOC inventories for
consumer products, but are extended to include materials/products used in commercial and
industrial processes (shown with [*] below):
• Paints (indoor and outdoor),
• Industrial adhesives,
• Consumer adhesives,
• Sealants,
• Pesticides (consumer and agricultural),
• General cleaning products,
• Targeted cleaners,
• Personal products (e.g. beauty and hygiene),
• Building materials,[*]



- Paving and roofing asphalt-related materials,

- Solid consumer products and packaging (e.g. plastics, elastomers), and their additives (e.g. plasticizers, flame retardants).[*]

However, the volatility range of most of the existing categories needs to be expanded to completely include I/SVOCs. We discuss I/SVOC-containing pesticides and asphalt-related products/processes as examples that are not sufficiently included in inventories (see Section 4.1.3). Notes: we exclude gasoline evaporation (from vehicles or fuel stations) and fossil fuel extraction/processing given their close connection to combustion. We briefly discuss food-derived cooking emissions for comparison, which are important and classified separately.

*Emission pathways:* We propose three general pathways of product/process-related emissions:

1) Evaporation of solvent from a product or during a process;

2) Volatilization of solute or active compounds from an applied coating or a solid product/material (i.e. off-gassing); and

3) Volatilization of by-products from the degradation or transformation of solute or active compounds.

The first is the principal pathway previously considered and often occurs over faster timescales (minutes-days) from liquids like cleaners, paints, and other solvents. The second acts over longer timescales (weeks-years), is not always included in inventories, and is a key potential source of I/SVOCs (see Section 4.1.3). SVOC pesticides, flame retardants, and off-gassing plastics are examples studied for indoor air quality (Batterman et al., 2009; Brodzik et al., 2014; Clausen et al., 2004; Faber et al., 2013; Kemmlein et al., 2003; Lewis, 2001; Wensing et al., 2005; Weschler





and Nazaroff, 2008). The third is due to the generation of degradation by-products via thermal or
photochemical stress, exposure to oxidants (i.e. OH, $O_3$, or $NO_3$), or other reactive chemical
processes. These emissions are largely understudied with the exception of asphalt-related
emissions and ozonation of indoor materials (Poppendieck et al., 2007a, 2007b; Salthammer and
Fuhrmann, 2007; Toftum et al., 2008).

*A lifecycle approach:* Emissions during application or immediate use are most commonly studied,
but many emissions related to products, process, and materials occur over much longer timescales.
In the future, emissions studies need to include all three potential emissions pathways across full
lifecycles of:
• Storage,
• Transport,
• Application,
• "Curing",
• Active use,
• Weathering,
• Restoration,
• Removal and disposal.
There are too many facets within the lifecycles of each source category to discuss in this paper,
but our case study on asphalt-related emissions in Section 4.1.3 demonstrates several aspects.
Production methods are not explicitly included here since they may fall under the purview of
industrial point sources, but with small, distributed production it may be appropriate to consider



the industrial/commercial processes as area-wide sources. Chemical transformations of materials
or products across their lifetimes need to be considered since they can influence the chemical
composition and rates of emissions. For example, transformations can occur during a wide variety
of in-use conditions or during the storage of unused or partially-used products/materials, both of
which can be over long storage periods in a variety of environmental conditions. Application
methodology is a determining factor in both emission rates and composition during application.
For example, common methods include direct liquid application or aerosolization, either via a
pressurized can or a nozzle and air compressor, and aerosolization provides a direct emission
pathway for all components and subsequent evaporation of aerosol droplets. Finally, emissions in
latter parts of lifecycles (i.e. weathering/deterioration, restoration, removal, and disposal) all need
to be considered.

**3. Methods**

We constrain the magnitude, chemical composition, and potential air quality impacts of
emissions using multiple bottom-up approaches: chemical characterization of a selection of
consumer products via gas chromatography-mass spectrometry and carbon isotope mass
spectrometry; a detailed assessment of emissions inventories and estimation of source
contributions to potential SOA and ozone formation; theoretical calculations of emission
timescales; and a survey of reported chemical composition in consumer product MSDSs. Details
on the emission timescale calculations can be found in Appendix B, and other additional
methods details can be found in the supporting information.



*Chemical speciation of consumer products:* We selected 17 consumer products across a range of
product types with attention to those with unresolved alkane or aromatic mixtures (Table 1). Top-
selling products from major commercial providers were selected in order to make a realistic
assessment of products that are currently in significant public use, and to avoid biasing the analysis
towards less-common products. With a limited sample size, only 1 water- and 1 ethanol-based
product were included. Diluted samples were prepared at a concentration of 1000 ng $\mu L^{-1}$ in either
hexane or toluene depending upon the product composition provided in their MSDS. The chemical
composition of the emittable fraction of products was determined using gas chromatography with
electron ionization-mass spectrometry (GC-EI-MS) (Agilent 7890B/5977). A 1μL syringe
(Hamilton) was used to directly inject the sample onto the GC column through an inlet held at 320
℃. During each injection, the column was initially held at 40 ℃ for 2 minutes, then ramped at 10
℃ $min^{-1}$ to 325 ℃ and lastly held at 325 ℃ for 5 minutes. Mass spectra of background subtracted
individual ion peaks were used to identify compounds in a sample using the NIST mass spectra
library. Calibration curves were drawn for five different concentrations of authentic standards
(AccuStandard) for diesel range alkanes ($C_{10}$-$C_{28}$), purgeable aromatics, and terpenes. The
emittable organic fraction of raw products is defined by volatility and approximated as VOCs +
IVOCs via our chemical analysis and reported MSDS data (Note: SVOCs excluded in this analysis
to focus on compounds that are fully partitioned to the gas-phase at equilibrium under typical
conditions). A carbon isotopic analysis was also carried out at the Keck Carbon Cycle Accelerator
Mass Spectrometer (AMS) facility at U.C. Irvine to measure carbon-14 (‰ $\Delta^{14}C$) and carbon-13
(‰ $d^{13}C$) relative to carbon-12 via a 0.5MV Compact AMS (National Electrostatics Corp.), and
estimate the fossil carbon content in individual products.



*Urban emissions inventories:* We used the California Air Resources Board (CARB) Almanac
emissions inventory and the U.S. EPA SPECIATE 4.4 source profiles to generate the total and
compound-specific emissions for our California and Los Angeles (Cox et al., 2013; U.S.
Environmental Protection Agency, 2014). The CARB emissions inventory is the most detailed
available with respect to source categories and basin-level rates. Los Angeles is our case study
megacity given its historical role in air quality research and policy with a multi-decadal record of
emissions data, ambient measurements, field campaigns, and publications (Bishop and Stedman,
2008; Fortin et al., 2005; Neligan, 1962; Warneke et al., 2012). In addition, the U.S. National
Emissions Inventory and Global Emissions Initiative (GEIA) inventories were used for nationwide
and worldwide comparison of VOC emissions respectively from solvents and on-road motor
vehicles.

We estimated total daily potential ozone and SOA formation in greater Los Angeles from the
product/process-related emissions included in the CARB inventory. Values were also calculated
for exhaust and evaporative emissions from on-road motor vehicles using CARB's EMFAC model
database and literature ozone and SOA yields for each source pathway and LEV generation
(Gentner et al., 2013, 2017; Zhao et al., 2017). Potential ozone formation values are based on
maximum ozone incremental reactivity (MOIR) values from the SAPRC-07 inventory (Carter,
2007; Gentner et al., 2013). The very diverse range of compound classes used in products,
materials, and processes remain largely understudied with respect to their SOA yields (e.g. esters,
siloxanes). Hence, literature SOA yields were used wherever possible (Algrim and Ziemann, 2016;
Chacon-Madrid et al., 2010; Chan et al., 2010; Gentner et al., 2012; Kwok and Atkinson, 1995;
Ng et al., 2006; Pankow and Asher, 2008; Sadezky et al., 2006; Tsimpidi et al., 2010), and



estimated for other unstudied compounds (Table S6). SOA yields are estimated at 10 µg OA m$^{-3}$
in urban, "high-NO$_X$" conditions (approx. >5 ppb). Ozone and SOA yields for each analyzed
consumer product and product categories are compared to other key sources. The SOA yields
provide a conservative, lower estimate of potential SOA without aqueous SOA despite studies
showing that aqueous pathways to SOA increase SOA yields for small oxidized precursors or their
oxidation by-products (Daumit et al., 2016; Jia and Xu, 2014). For the case study city Los Angeles
(and Mexico City), aqueous SOA formation was relatively small during major field studies
(Dzepina et al., 2009; Hayes et al., 2015; Washenfelder et al., 2011). However, future work in
other cities should consider aqueous SOA given the magnitude of oxygenated fraction in
product/process-related emissions (see section 4.1.2).

*Survey of material safety datasheets (MSDS):* We obtained the MSDS data by surveying a set of
88 MSDS entries from the websites of major home improvement stores focusing on their top-
selling products. Five product categories were chosen including paints, adhesives, cleaning
products, sealants and pesticides. Chemical composition information was extracted from the
"composition/information on ingredients" section reported in the datasheets. MSDS entries for
commercial products frequently report 30% to 60% of product composition as "proprietary
mixtures", so this survey only identifies the presence and establishes general ranges for current
product types (Table 2).

**4. Results and Discussion**
**4.1 Composition and magnitude of product/process-related emissions**



The organic composition of consumer products and commercial/industrial products, materials, and
processes are very diverse, which leads to similar diversity in emissions, and further region/nation-
specific heterogeneity can be expected. We calculate the magnitude and average chemical
composition of emissions from source categories included in the CARB inventory (Figure 2a) with
the goal of assessing the distribution of emissions across organic compound classes and product
types (i.e. source profiles) in a megacity with the most representative inventory available. We find
that the consumer products and commercial/industrial processes that comprise product/process-
related sources are large emitters of a diverse suite of VOCs, but view these results as a lower
estimate given likely missing emissions, such as those discussed in Section 4.

### 4.1.1 Chemical composition

*Laboratory analysis of consumer products:* Our results summarized in Table 1 demonstrate the
prevalence of non-benzene, single-ring aromatics and $C_6$-$C_{12}$ alkane mixtures as solvents, and the
presence of IVOCs and SVOCs in consumer products. The emittable fraction of products ranged
3-100%, and the single-ring aromatic and IVOC content ranged 3.5-93% and 0.77-95%,
respectively. Another key result of this analysis is the frequency of many functionalized aliphatic
or aromatic VOCs and IVOCs that are not traditionally measured in atmospheric monitoring (e.g.
esters, acetates, siloxanes), which is echoed in the MSDS survey results. More detailed speciation
results with a breakdown of alkanes and single-ring aromatic compounds can be found in Table
S1. Validation of real-world product/process-related emission rates/timescales is necessary to
advance the field but will require examining a wide range of products and geographic conditions.



*Analysis of emissions inventories:* Figure 3(a) shows the estimated overall composition of
product/process-related emissions in Los Angeles averaged over the years 2005 to 2020
determined by combining SPECIATE source profiles with the CARB emissions inventory for the
South Coast Air Basin (SoCAB). Alcohols and miscellaneous emissions together make up 50% of
the total product/process-related emissions with ethanol as one of the largest individual species
emitted, which is consistent with SoCAB ethanol observations that were substantially greater than
what would be expected with gasoline vehicles as the only source (de Gouw et al., 2012). 70% of
the miscellaneous emissions are made up of mineral spirits whose composition varies greatly with
application but are generally comprised of acyclic and cyclic $C_{7-12}$ alkanes with variable amounts
of aromatic content. The remaining 30% includes unresolved asphalt mixtures, oxygenates,
fragrances, and undefined petroleum distillates/oils/spirits, some of which also fall into the
I/SVOC category discussed below. Single-ring aromatics are estimated at 13% of total emissions
with a mix of 43% toluene, 38% $C_8$, 3% $C_9$, and 1.5% $C_{10}$ aromatics, and minor PAH emissions.
Carbonyls represent 10% of total emissions, 36% of which is acetone. Anthropogenic terpenoid
emissions in the inventory, while highly reactive, are small relative to other sources and biogenic
contributions (see section S.4 in SI).

Figure 3(b) shows the distribution of organics in the SoCAB emissions inventory from major
product/process-related source categories. Paints emerged as the highest VOC emitter, ~21% of
which is single-ring aromatics (Figure S4). Targeted cleaners, pesticides, and general cleaners
were the next largest source types in the inventory. A more detailed breakdown of emissions
broken up by source category and compound class can be found in Table S3.



*MSDS survey:* Summarized in Table 2, the greatest single-ring aromatic content was observed in
adhesives (8%) and sealants (9%), while the 30 surveyed paints did not have aromatic VOC content
in contrast to the SPECIATE source profiles. In comparison, the chemical speciation of 42 major
solvents (across $C_{6-13}$) in 2002 reports a wide range of total aromatic content (0-100%) and an
average of 41% (±46%) with the remainder comprised of acyclic and cyclic alkanes (Censullo et
al., 2002). Differences when compared to our present-day laboratory and MSDS paint speciation
show reductions in aromatic content due to increasingly stringent regulations. However, for other
locations, it highlights the likely, continued prevalence of single-ring aromatics in solvents,
especially for developing regions. We also found that MSDSs did not report I/SVOCs content in
the composition of pesticides while our laboratory speciation found a 8%-95% IVOC content in
the emittable fraction of the analyzed pesticides samples. For all of the product categories
examined, and especially for pesticides, the amount of compositional information provided by the
MSDSs is limited and hinders our ability to constrain the average aromatic or I/SVOC content
since compounds outside the VOC range are often not disclosed due to proprietary claims or
regulatory exemptions, and where provide all compound concentrations are usually provided as a
wide range. For example, under "fragrance exemptions" VOCs and IVOCs are often not disclosed
and can even be labeled as "VOC-free".

### 330     4.1.2 Emission rates

While motor vehicles are still major sources in developed regions, their total gas-phase organic
compound emissions have been gradually declining with the continued implementation of stricter
emissions standards (Gentner et al., 2017), increasing the relative importance of other sources
(Figures 1, 2a). At a global scale, the MACCity and ACCMIP emissions inventories estimate





global solvent-related VOC emissions of 15 Tg yr$^{-1}$ in 2000 and year-over-year increases since
1960 (speciated by aromatics, C$_{6+}$ alkanes, ketones, alcohols, and other VOCs), and emissions of
aromatics from solvents are expected to outweigh those from transportation in 2020 (7.5 vs. 6.7
Tg yr$^{-1}$) (GEIA, 2017). The 2014 U.S. National Emissions Inventory (NEI) data reports that VOC
emissions from solvent-related sources are just 25% less (300 tons day$^{-1}$) than those from on-road
mobile sources nationally, while they exceed on-road emissions by 25% (21 tons day$^{-1}$) in
California (U.S. Environmental Protection Agency, 2011). Figure S1 shows California's statewide
bi-decadal emissions estimates from CARB where product/process-related VOCs will reach 450
tons day$^{-1}$ in 2020, exceeding motor vehicles by 116 tons day$^{-1}$. A similar picture may emerge in
developing nations, as the control of motor vehicle emissions is greatly accelerated by the
advancements and knowledge of developed nations; such that emission standards in major
emerging economies employ either U.S. or E.U. policies, and are generally only one generation
behind (Kodjak, 2015). International studies over the past 2 decades show highly varying
contributions (~5-45%) to the total anthropogenic VOC emissions from just "solvent" use at both
regional and national scales (summarized in SI) (van den Born et al., 1991; Caserini et al., 2004;
Chen et al., 2009; Deguillaume et al., 2008; Lu et al., 2007; Markakis et al., 2009; Menut, 2003;
Nielsen et al., 2008; Piccot et al., 1992; Song et al., 2007; U.S. Environmental Protection Agency
and Office of Air Quality Planning and Standards, 1991).

Emissions inventory data for Los Angeles demonstrate that anthropogenic emissions have
consistently decreased over the last four decades (Figure 2a), which is consistent with ambient
observations (1960-2010) (Warneke et al., 2012). As the contribution of on-road motor vehicles
to total anthropogenic emissions has declined, product/process-related sources have become a



major contributor of VOCs. Based on the CARB emissions inventory, contributions of
product/process-related sources are the largest single contributor of VOC emissions in the basin
and state (Figures 2a & S1), and are growing with population (i.e. increased in-basin usage). While
consumption volume is low compared to combustion fuels, emission factors are higher given that
most of the volatile components are emitted whereas fuels are burnt at ≥ 99% efficiency. Yet, no
region can be fully representative of product/process-related emissions on a larger scale, and
regional/national specifics will influence the magnitude and composition of emissions. For
example, California's extensive regulations will modify the composition of products sold in CA,
and likely elsewhere in the U.S.

**4.1.3 Emissions of intermediate- and semi-volatile organic compounds (IVOCs and SVOCs)**
Despite representing only a small to moderate amount of emissions, IVOCs and SVOCs from
motor vehicles are key precursors to urban SOA (Gentner et al., 2012; Robinson et al., 2007).
Similarly, we conclude that consumer products and commercial/industrial processes are also large
sources of unspeciated IVOCs and SVOCs, some of which are included in the CARB inventory,
while there is evidence that some other source pathways are not.

Many common products/materials (e.g. pesticides, fragrances, foams, plastics) have IVOCs or
SVOCs that partition to reach equilibrium and evaporate over long timescales (Figure 4)
(Batterman et al., 2009; Clausen et al., 2004; Mitro et al., 2016; Weschler and Nazaroff, 2010).
Our chemical analysis revealed IVOCs or SVOCs in 10 of the 17 products (Tables 1 & S1).
Including the MSDS survey, we found composition ranging from 0% to 95% I/SVOCs, with



I/SVOCs present in 23% of cleaners, 20% of adhesives, 24% of paints, and 46% of sealants (Table
2). Aliphatic or aromatic I/SVOCs are frequently used in some types of pesticides and are
sometimes replaced with biogenic oils (e.g. neem oil, fish oil), such that the bulk of the pesticides
in our MSDS survey were comprised of < 1 - 10% active compounds and a balance of undisclosed,
"non-hazardous" ingredients.

The SPECIATE profiles and CARB emissions inventory include some estimates of unspeciated
"low-vapor pressure VOCs (LVP-VOCs)" in consumer products that are defined as larger than 12
carbon atoms (or equivalent volatility), which is roughly consistent with the beginning of the IVOC
range (California Air Resources Board, 2015b). While poorly constrained, non-aromatic I/SVOCs
included in the inventory constitute ~3% (6 tons day$^{-1}$) of total SoCAB emissions estimates in the
CARB/SPECIATE case study, 90% of which is classified as unspeciated "LVP-VOCs". Consumer
product pesticides, general purpose cleaners, and targeted cleaners (e.g. laundry) are the largest
sources of I/SVOC contributions emitting 3.6, 1.0 and 0.4 tons day$^{-1}$ of I/SVOCs, respectively.
Consumer product pesticide emissions had the highest fraction of I/SVOCs (29%).

While CARB's consumer products inventory includes I/SVOCs due to manufacturer reporting
requirements, I/SVOCs are exempt from limits on VOC content except for multipurpose solvents
and paint thinners, and there are known limitations in their coverage, especially of oxygenated
species (California Air Resources Board, 2000a, 2015b). We conclude there are other
anthropogenic sources of IVOCs and SVOCs that have not been considered due to their long
emission timescales (i.e. days-years; Figure 4, Table S9) (de Gouw et al., 2011; Weschler and


Nazaroff, 2008). We present evidence for two such examples in this paper: I/SVOC-containing
pesticides and asphalt-related products, materials, and processes, but other examples include the
volatilization of I/SVOC solvents, solutes, or solids (e.g. coatings, flame retardants) and materials
that may degrade to form compounds with volatilities of $C_{13-26}$ n-alkanes (e.g. construction
materials/coatings, including materials with petroleum distillates/residues, mineral oil, coal tar, or
similar).

***Example 1: Emissions of IVOCs and SVOCs from pesticides:*** We have several pieces of evidence
that demonstrate pesticides (including herbicides, insecticides, and fungicides) contain I/SVOCs,
but they also highlight the fact they are poorly documented and regulated. We analyzed the
chemical composition of three pesticides available as consumer products and they were comprised
of 8%, 20% and 95%, IVOCs in the $C_{14-17}$ range (Table 1, Figure S3), with trace levels of larger
compounds. Our MSDS survey (Table 2) was inconclusive for pesticides since the majority of
consumer pesticides are not disclosed in MSDSs due to claims of proprietary mixtures, non-
regulated components outside of the VOC range, and/or because they use naturally-derived oils
(e.g. neem oil). Studies have shown such naturally-derived oils are comprised of aliphatic and
aromatic I/SVOC-range compounds (Isman, 2000; Kumar and Parmar, 19996). Our emissions
inventory analysis shows that non-aromatic I/SVOCs are the largest inventoried source of
I/SVOCs (Figure 3), but the lack of data in MSDSs and VOC exemptions from regulations suggest
this is a lower estimate of actual emissions. To examine commercial/industrial products, we also
analyzed the composition of three petroleum-based pesticides used in agriculture (Figure S2),
which are comprised of $C_{16}$-$C_{26}$ cyclic and acyclic alkanes mostly in the SVOC range that are
applied as thin films via a sprayed water emulsion. These can partition to the gas phase and be re-



emitted on the timescales in Figure 4 and may impact urban (or downwind) areas in agricultural
regions, with 18-29 Gg applied yr$^{-1}$ in California.

*Example 2: Emissions of IVOCs and SVOCs from asphalt-related products, materials, and*
*processes:* We propose asphalt-related products/processes as important sources of IVOCs and
SVOCs whose emissions are currently underestimated in inventories and require better
quantification. Asphalt-containing materials are used in road paving and repair (and similar
applications for roofing or other surfaces), and are comprised of petroleum-derived organic
compounds; predominantly non-distillable (i.e. non-volatile) asphalts sometimes with smaller
amounts of VOCs, IVOCs, and/or SVOCs. They are used as sealers, coatings, and binders; mixed
with aggregates to pave roads; and applied using either high application temperatures, water
emulsions, and/or solvent.

The three paving/roofing-related products we analyzed contained aliphatic and aromatic VOCs
and IVOCs up to $C_{18}$ present as solvents, with minor SVOC content (Figure S3). Similarly, there
were no I/SVOCs declared in the asphalt-containing products in our MSDS survey. Non-solvent
emissions during the hot storage, application, or resurfacing of these asphalts are caused by the
degradation (i.e. fragmentation) of larger asphalts to form smaller compounds ($C_7$-$C_{30}$), which
include cyclic and acyclic alkanes, single-ring aromatics, PAHs (2-, 3-, and 4-ring), and sulfur- or
nitrogen-containing species (i.e. benzo- and dibenzo- thiophenes and furans), all of which were
not present in the asphalt prior to heating (Cavallari et al., 2012b; Gasthauer et al., 2008; Kitto et
al., 1997; Kriech et al., 2002; Lange et al., 2005; Lange and Stroup-Gardiner, 2007; The Asphalt



Institute & European Bitumen Association, 2015). The total mass and composition of emissions is
dependent on production methods, asphalt grade, and increases in magnitude and maximum
molecular weight with storage/application temperatures (i.e. more SVOCs at high temperatures),
which ranged 100-240 ºC, or higher for roofing asphalts (Cavallari et al., 2012b; Gasthauer et al.,
2008; Kitto et al., 1997; Kriech et al., 2002; Lange et al., 2005; Lange and Stroup-Gardiner, 2007;
The Asphalt Institute & European Bitumen Association, 2015).

Emissions of aromatic and aliphatic VOCs, IVOCs, and SVOCs from heated asphalt mixtures
("hot mix") during application have also been documented in occupational health studies on
"asphalt fumes" (Cavallari et al., 2012a, 2012b; Kriech et al., 2002; Lange et al., 2005; Lange and
Stroup-Gardiner, 2007). Yet, current emissions inventories do not include emissions of VOCs,
IVOCs, and SVOCs from the degradation of larger compounds during and after the application of
asphalt mixes. Estimation methods focus solely on the evaporation of VOC solvents from "cutback
asphalt", included as an area source in the "solvent evaporation" category in Californian, U.S., and
E.U. inventories (California Air Resources Board and Sonoma Technology Inc., 2003; San Joaquin
Valley Air Pollution Control District, 2008; U.S. Environmental Protection Agency, 2014; U.S.
EPA, 1995). Road paving solvents are prohibited in non-attainment areas in California (Table S4),
so emissions in the SoCAB case study are minor (1 ton day$^{-1}$) and mostly smaller than $C_{10}$ in the
SPECIATE source profiles (Table S5) (Cox et al., 2013).

Emission factors of degradation byproducts don't exist, so we approximate lower limits on
emission factors only for the period immediately during application using limited published data





(see section S.3 in SI), but longer timescale experiments are necessary. Calculated lower limits
range from 100-2000 mg kg$^{-1}$ of asphalt (not including aggregate) with a strong dependence on
application/storage temperature. This is on the same order as motor vehicle emission factors and
is greater than CARB's current emission factor for hot-mix asphalt (District, 2012; Gentner et al.,
2017; San Joaquin Valley Air Pollution Control District, 2008). Yet, California's asphalt
consumption of 1,540,000 tons liquid asphalt year$^{-1}$ (Table S4) represents statewide I/SVOC
(+VOC) emissions of 0.5 – 8 tons day$^{-1}$ during application alone (The Asphalt Institute, 2015).
This does not overwhelm current solvent-VOC emissions from paving/roofing (33 tons day$^{-1}$) but
emphasizes the need for further research since the poorly-constrained emissions largely include
I/SVOCs emitted over long timescales which are known to have high SOA yields.

Asphalt-related emissions exemplify the stated need for lifecycle-focused approaches, with
potential emissions across storage, transport, application, curing, weathering (e.g. degradation due
to climate or UV radiation), and resurfacing. Their magnitude and composition will vary with
production/handling methods, geologic source, and application type and location (esp. climate).
Emission pathways (from Section 2) include (1) volatilization of application solvents and (3) the
production and release of degradation byproducts while (2) does not apply due to negligible off-
gassing from extremely low volatile un-degraded asphalt constituents.  The emission of asphalt
degradation products may peak during construction-related activities primarily due to asphalt's
exposure to high temperatures during its storage, application, or resurfacing. Still, seasonal highs
in surface temperature (summer pavement maximums are 47-67 °C and up to 70 °C for roofs
(Parker et al., 1997; Pomerantz et al., 2000)), will likely affect the rate of internal transport and
diffusion out of the "cured" asphalt layer resulting in emissions extended over its lifetime. (Note:



paving solvents are currently assumed to be emitted over several months (California Air Resources
Board and Sonoma Technology Inc., 2003)).

**4.2 Potential SOA and ozone formation of product/process-related emissions compared to**
**other major sources**
Products and processes emit a diverse array of organic compounds (Figure 3). Some are of low
direct concern for human health (terpenoids, siloxanes etc.), while others present issues as primary
emissions, especially in indoor or concentrated workplace environments (aromatics, ethers,
PAHs). Yet, most are reactive and will oxidize in outdoor or indoor environments to form oxidized
VOCs with unknown, but large potential health effects (Pöschl and Shiraiwa, 2015). This section
focuses on their impacts on air quality via SOA or ozone formation. However, the health or
environmental effects of the primary emissions should be especially considered in developing
regions where primary VOC emissions are larger, or for specific compound classes near sensitive
natural environments.

A comparison of the SOA yields and ozone formation potentials for major source categories
(Figure 5) demonstrates that product/process-related emissions have SOA yields and ozone
formation potentials that are on par with other major urban sources such as motor vehicles and are
strongly dependent on composition. Gas-phase cooking emissions from food represent an
additional uncertain source in urban air quality along with cooking POA which has been more
studied (Bruns et al., 2017; Hayes et al., 2013; Klein et al., 2016).



*SOA formation potential:* The potential SOA from on-road gasoline vehicles in greater Los
Angeles region has decreased by ~65% between the years 1990 and 2015. By 2020, further
reduction by 25% is expected relative to the 2015 value of 3.3 tons day$^{-1}$. These numbers for diesel
vehicles are 75% and 1.3%, respectively. If the emissions inventory is accurate, then the 2015
potential pp-SOA in the SoCAB basin is nearly equal to the SOA formation potential of on-road
gasoline and diesel vehicles, and is estimated to surpass them by 2020 with an increasing share of
SoCAB's total anthropogenic emissions (Figure 2b). While single-ring aromatics and PAHs
constitute only 11% of the total product/process-related emissions, they are responsible for ~80%
of the potential SOA from those sources in the region (4% PAHs and 76% single-ring aromatics,
largely toluene and xylenes). Existing emission inventories badly underestimate emissions of
I/SVOCs and their contributions to SOA. They are shown to be responsible for 0.18% of pp-SOA,
but this excludes 87% of the total I/SVOCs emissions which are labeled as unspeciated "LVP-
VOCs" in the inventory and thus have no assigned SOA yield in calculations. Additionally, our
calculations do not include "missing" emissions or their potential SOA. Anthropogenic terpenoids,
including lemon oil, pine oil, orange oil, orange terpenes, D-limonene and α-pinene, are
responsible for ~8% of the total product/process-related SOA.

*Ozone formation potential:* Over the past several decades, potential ozone has been dominated by
emissions from gasoline motor vehicles. Yet, in 2015 potential ozone from gasoline vehicles was
only 30% greater than product/process-related sources, and by 2020 product/process-related
emissions will surpass on-road motor vehicle contributions in the basin (Figure 2c). Potential
ozone from on-road diesel vehicles is only ~5% of that from product/process-related sources.
Contributions to potential ozone from product/process-related sources are 33% alcohols, 29%



aromatics, and 12% alkanes (not including unspeciated "LVP-VOCs") (Table S2). A recent ozone
formation sensitivity analysis of solvent-related emissions speciation with 22 lumped species
demonstrated that using input source profiles that are more detailed in terms of contributing
compound classes would improve ozone model performance (von Schneidemesser et al., 2016).

*Contributions from off-road combustion-related sources to potential SOA and ozone:* Off-road
mobile sources are also significant sources of reactive precursors to SOA and ozone across a very
diverse mix of vehicles, boats, equipment, and other engines, and sometimes operate on specialized
fuels (e.g. aviation gasoline, jet fuel, jet naphtha, fuel oil) (Cox et al., 2013; Gordon et al., 2013;
May et al., 2014; Zhao et al., 2016). These sources have received greater regulatory attention in
the past 20 years, and control policies (e.g. CARB, EPA) will add additional variance to the
magnitude and composition of gas-phase organic compounds they emit (California Air Resources
Board, 2017; Miller and Facanha, 2014; U.S. EPA, 2017). Given the limited information on their
diverse source profiles, and thus their SOA and ozone yields (Gordon et al., 2013; McDonald et
al., 2015), we constrain their uncertain contributions independent from on-road sources. Their
fraction of anthropogenic VOC emissions in the SoCAB leveled off after 2005 and started to
decrease (Figure 2a), and in 2015 the ratio of on- to off-road emissions in the ARB inventory
almanac is 1.5:1 and 1:2 for gasoline- and diesel-powered engines, respectively, when including
equipment, recreational vehicles, boats, trains, and aircraft that use either gasoline or diesel (Cox
et al., 2013). So, given similar ozone and SOA yields to on-road gasoline or diesel vehicles, off-
road emissions could approximately increase potential SOA and ozone contributions from
gasoline-related sources by 67% and diesel-related sources by 200% for the year 2015 in the
SoCAB, but are subject to the uncertainty from the wide range of engines, fuels, and emissions



controls affecting the composition of emissions. These results are generally consistent with the
relative source contributions of on- and off-road sources to total OA reported by McDonald et al.
(2015) for 2010 in the SoCAB. The current outsize impact of off-road sources, despite using a
relatively small amount of fuels, is due to the fact that emission factors are much higher for off-
road sources, such as 2+ orders of magnitude higher for gasoline off-road compared to on-road
sources (Gordon et al., 2013; McDonald et al., 2015; Zhao et al., 2016).

The inclusion of off-road sources does not affect our conclusion in this section that non-
combustion sources and motor vehicles (on- and off-road) contribute similar amounts of potential
SOA and ozone in our case study megacity around the 2015-2020 period, with a rapidly growing
role for non-combustion sources as combustion emissions are further controlled. In all, this
highlights the importance of continued assessment and regulation of off-road combustion-related
sources as part of a holistic air quality management plan along with non-combustion sources and
on-road vehicles.

***Modifying factors for SOA and ozone formation chemistry:*** Relative VOC to $NO_X$ ratios have
been shown to affect the chemistry and production rates of SOA and ozone (Hallquist et al., 2009;
Sillman, 1999; Zhao et al., 2017). For many cities outside Los Angeles, urban air quality develops
within a backdrop of biogenic emissions of VOCs and IVOCs, which is critical to keep in mind as
we pursue stricter emission targets for reactive organics. In some cases, such as in the Southeast
U.S., biogenic emissions dominate over anthropogenic gas-phase organics, and emissions of $SO_2$
and NOx are large drivers of biogenic SOA formation (Xu et al., 2015).




### 4.3 Empirical ambient evidence for IVOC and SVOC emissions, and their SOA contributions

Ambient measurements (2010) of I/SVOCs in our case study city are consistent with our findings;
they demonstrate that IVOCs are important contributors to SOA in the region and other urban
areas, but major uncertainties persist regarding the sources of primary IVOC emissions (Hayes et
al., 2015; Ma et al., 2016; Zhao et al., 2014). Recent model results estimate that 70-86% of urban
SOA in Pasadena come from the oxidation of primary I/SVOC emissions (Ma et al., 2016). Zhao
et al. (2014) state that unidentified non-vehicular sources contribute 'substantially' to these
emissions but no clear fraction is yet established. A major fraction of SOA cannot be explained
without the inclusion of IVOCs or other unspeciated organics (Gentner et al., 2012, 2017; Hayes
et al., 2015; Jathar et al., 2014; Zhao et al., 2014). Other results indicate that fossil-related sources
contribute approximately half of OA and a majority of fresh, urban SOA in Los Angeles (Gentner
et al., 2017; Hayes et al., 2013; Zotter et al., 2014). Yet, bottom-up estimates and top-down
assessments of the SOA produced from gasoline and diesel vehicles (on- and off-road) cannot
explain all of the observed fossil SOA in LA in 2010, which supports our conclusions and suggests
the presence other major sources of fossil-derived SOA precursors (Ensberg et al., 2014; Gentner
et al., 2017; Ma et al., 2016; McDonald et al., 2015; Zhao et al., 2014).

### 4.4 Isotopic carbon content and interpreting ambient isotopic data

The isotopic carbon content (i.e. $^{14}C$ vs. $^{13}C$ vs. $^{12}C$) of organic aerosol has been used directly, and
in tandem with source apportionment of bulk aerosol data from an aerosol mass spectrometer, to
infer fossil vs. non-fossil origin of carbonaceous aerosols and their SOA precursors at several





locations (Ceburnis et al., 2011; Hayes et al., 2013; Zotter et al., 2014). We tested the potential
effect of their emissions on the interpretation of isotopic measurements of ambient SOA, and our
isotopic analysis of 12 products demonstrates that their VOC and I/SVOC emissions and thus SOA
will be depleted in Carbon-14 (Table 1). 8 of the 12 contained 97% or more fossil carbon, while
the remaining 4 contained 57-81% fossil carbon. Asphalt-related sealants, solvents (e.g. paint
thinner, naphtha), and pesticides were found to have the highest fossil carbon content. This is
consistent with the fact that petrochemical feedstocks derived from petroleum and other fossil fuels
are used in the products, materials, and processes discussed throughout this work.

Studies in greater Los Angeles (Pasadena) report that fossil-fuel driven emissions appear to
contribute 68%-74% of the observed afternoon increase in SOA formed from urban sources with
the remaining ~25% coming from non-fossil sources, principally regional biogenic and local
cooking sources (Hayes et al., 2013, 2015; Zotter et al., 2014). The observed fossil SOA and its
potential precursor sources have been analyzed across several studies with differing conclusions
(Bahreini et al., 2012; Gentner et al., 2017; Hayes et al., 2013; Zotter et al., 2014). The studies
agree that on-road diesel vehicles are a relatively minor contributor to fossil SOA (Bahreini et al.,
2012; Hayes et al., 2013, 2015; Zotter et al., 2014), and along with other papers, have highlighted
the importance of other anthropogenic sources (Ensberg et al., 2014; McDonald et al., 2015; Zhao
et al., 2014). Other evidence suggests a fossil source other than on-road diesel since concentrations
of IVOCs show only minor weekday vs. weekend variation with changes in diesel traffic (Zhao et
al., 2014). Based on our results and the evidence in the literature, we conclude that fossil-derived
urban SOA precursors are emitted from product/process-related sources, and are responsible for
some of the fossil SOA observed in the SoCAB. The isotopic signature of products, materials, and



processes is due to their petrochemical feedstocks, and this has led to their misattribution to
combustion-related sources in the past.

**5. Conclusions and future research needs**
Using multiple bottom-up approaches, we demonstrate the growing importance of non-combustion
emissions of gas-phase organic compounds from anthropogenic sources. Yet, our understanding
has been inhibited due to the chemical diversity of emissions across a myriad of source types, their
fossil isotopic signatures, and in many cases their prolonged emission timescales, which occur
over full lifecycles and a broader range of emissions pathways than is typically considered.
Emission timescales can extend over months or longer (Figure 4) in the case of: thick layers of
materials/coatings; sources of I/SVOCs, or the formation and emission of degradation by-products.
The implications of these prolonged timescales are a legacy of unreleased potential emissions built
up or "banked" in the products and materials spread across urban areas.

It is critical to emphasize that these results do not justify deviating attention from, or relaxing
emission standards for, combustion-related sources since they are still prominent factors in urban
air quality in the developed and developing world, and remain dominant contributors of carbon
dioxide. For the foreseeable future in many locations, they remain principal contributors of reactive
organic precursors and other criteria pollutants, especially in near-source hotspots, such as
roadways. Rather, we conclude that in order to support the coming decades of policy, modern air
quality research needs to holistically consider the full portfolio of anthropogenic (and biogenic)
sources that impact urban air quality. We highlight key research needs to support this objective.




***A broad perspective on non-traditional sources of reactive carbon:*** This analysis highlights the importance and further consideration of VOC, IVOC, and SVOC sources that have not received sufficient attention to effectively support policy. With the successful control of "low-hanging fruit" where single source types dominated emissions, the control of a broader array of disparate sources becomes necessary. It is likely that there are other non-vehicular sources of reactive gas-phase organics in urban areas (e.g. food-derived cooking emissions), which need to be better included in inventories and models. Monitoring, studies, and inventories need to comprehensively include functionalized compound classes rather than just traditional classes.

658

***Validation of existing emissions inventories:*** Our results based on exiting emissions inventories are subject to uncertainties in their methodology. Given the results, further research is needed to review, evaluate, and validate both the emissions factors and source profiles in emissions inventories. A lifecycle approach should include lifetime emissions over the three pathways in Section 2. Such emissions will be seasonally-dependent on factors such as air velocity, relative humidity, and temperature (Wolkoff, 1998). A large survey of products and materials with attention to national/regional differences is warranted to ensure that source profiles accurately represent product/process-related emissions. Single-ring aromatics are a key example; the current CARB inventory using the SPECIATE source profiles reports large emissions of single-ring aromatics, 72% of which comes from paints. Yet, this is in contrast to our MSDS survey that observed little aromatic content in current-day U.S. paints (Table 2).




***Inclusion of IVOCs and SVOCs in emissions inventories:*** There is a clear need for more detailed
emissions inventories of I/SVOCs from all sources. Further research should support the evaluation
of the I/SVOC exemption in strategic air quality management plans and composition reporting
requirements in MSDSs and similar databases. More detailed speciation is required to accurately
determine the SOA (and ozone) formation potential of unresolved I/SVOC mixtures. Emerging
measurement methods will enable a more robust update of the currently unresolved I/SVOCs in
existing source profiles, which sometimes stem from outdated survey data (California Air
Resources Board, 2000b). In the process, attention should be paid to heterogeneity in products
containing petroleum distillates and similar components, which are very broad and can result in an
equally broad range of emissions.

The off-gassing of I/SVOCs (and VOCs) from materials was not experimentally tested in this study
and is not included in inventories. Yet, recent studies have shown plastics, foams, and building
materials off-gas I/SVOCs and VOCs, such as aromatics, aliphatics, halocarbons, terpenes,
organophosphates, and oxygenated species (Brodzik et al., 2014; Faber et al., 2013; Kemmlein et
al., 2003; Toftum et al., 2008; Wensing et al., 2005). Similarly, detailed lab and field
characterization of asphalt-related emissions over long timescales is needed to constrain emission
factors, source profiles, and the effect of modified asphalts (e.g. Superpave).

Understanding the application method and environment is key to determining air emissions of
I/SVOCs since some products may be used with water and disposed of into wastewater, described
as *down-the-drain factors*. For example, a recent ozone modeling study modeled the fate of 23





oxygenated I/SVOCs (e.g. glycols, glycol ethers, esters, alcohols) present in cleaning and personal
products that are used with water found that most of the compounds that go down-the-drain do not
volatize and are biodegraded at wastewater treatment plants, while >90% of compounds that
volatize outdoors will react with OH and contribute to ozone formation (Shin et al., 2015).

*Quantifying outdoor transport of indoor emissions:* Similar to *down-the-drain* factors, we
highlight the need for a similar factor to determine fractions of indoor emissions that are
permanently lost to indoor sinks via chemical or physical deposition (*fraction lost to indoor sinks*).
Similarly, models will need factors to account for the increase in characteristic emission timescales
when products/materials used on indoor surfaces remain indoors longer due to generally lower
vertical transport coefficients and subsequent re-partitioning other surfaces prior to transport
outdoors.

*Inter-location variability in the developed and developing world:* While we present annual trends
for our case study megacity Los Angeles, a similar situation is evolving elsewhere in developed
urban areas where emissions of anthropogenic organics are key drivers of SOA and ozone
formation, especially in megacities. We expect substantial heterogeneity between locations in the
composition of emissions from products, materials, and processes due to national/regional
regulations governing formulation, as well as climate, application specifics, and consumer
preferences and options. In particular for this work, California's advanced regulatory program may
have led to the phase out of some components in U.S. products. So globally the composition and
magnitude of product/process-related emissions may contain a much greater fraction of reactive



species, such as single-ring aromatics. Top-down ambient studies and bottom-up studies for other
locations are needed to confirm the importance of product/process-related emissions, and air
quality modeling studies should support these efforts. The situation in urban areas of developing
regions and emerging economies is uncertain since they have motor vehicle emissions
standards/technologies that may be much more advanced than the rest of their air quality
management plans (Kodjak, 2015). In such locations, non-combustion sources may be important
sources, but the combustion of fossil fuels or biomass for home heating/cooking and agricultural
waste disposal will play a larger role than in developed regions. In all locations, the magnitude of
regional biogenic emissions and $SO_2$/$NO_X$ emissions will affect the impact of anthropogenic
organics on SOA.

*Understudied oxidation pathways and products:* The oxidation pathways and products for many
of the functionalized compound classes discussed in this work (e.g. Figure 3) are largely
understudied, with a lack of experimental or theoretical studies to constrain the generation of
SOA, ozone, and oxidized gases. Such oxidation products/pathways are also particularly
important in indoor environments, so research on their precursor emissions and subsequent
oxidation is also important for indoor air chemistry.

**Acknowledgements**
For the agricultural pesticide analysis, we would like to thank John Karlik (U. California
Cooperative Extension Kern County) for samples, Allen Goldstein (UC Berkeley) for access to
analytical instrumentation, and Emily Barnes (Yale) for help with preliminary analysis. We also



thank Jonathan Williams (Max Planck Institute for Chemistry) for his feedback on the
manuscript and the reviewers whose comments helped improve this manuscript.

**Supporting Information.** Please see the supplemental material for additional details on relevant
regulations, methods, Tables S1-S9, and Figures S1-S4.

**Appendix A: Current and historical regulations and policy on non-combustion products**
**and processes**

Emissions from consumer products and industrial processes received some attention in pre-2000
outdoor air quality research and policy, especially in the cases of toxic components, highly-reactive
volatile solvents that fueled rapid ozone production (e.g. alkenes), and stratospheric ozone-
depleting chemicals. Emissions of a select few hazardous air pollutants (HAPs) (e.g. benzene and
vinyl chloride) were first broadly regulated under the U.S. Clean Air Act (1970). Subsequent
amendments through 1990 required the U.S. Environmental Protection Agency (EPA) to regulate
key sources of precursors to ozone production and emissions of 189 newly-designated HAPs (now
"air toxics"), some of which were used in products and processes (National Research Council,
2004). Less well-known, the 1990 amendments also required the EPA to identify priorities and
guidelines to mitigate emissions from consumer and commercial products (National Research
Council, 2004). Several categories of paints and solvents were subsequently identified based on
results from paint drying and chamber experiment studies conducted during early 1970s through
late 1990s (Chang et al., 1997; Clausen et al., 1993, 1990, 1991; Hansen, 1974; Sparks et al., 1999;
Sullivan, 1975), and regulations were established for half of the product categories, but actions
were ultimately halted as these sources were not viewed as central to ozone or other criteria



pollutant mitigation at the time (National Research Council, 2004). Despite large uncertainties
about emissions and ambient contributions from products/processes, this strategy reflects the
magnitude and impact of motor vehicle emissions in 1990 (Figures 1-2); the keen focus on ozone
production; and the lack of knowledge on SOA formation chemistry and I/SVOCs.

U.S. state or air basin-level regulations vary with region and attainment status. California's air
quality policy and regulations have been the most inclusive and detailed with respect to emissions
from consumer products and some industrial processes. The California Air Resources Board
(CARB) started the Consumer Products Regulatory Program in 1991, and a similar Coatings
Program, to address outdoor and indoor air quality problems associated with their emissions by
placing product type-specific limits on VOC content, and total reactivity limits (i.e. ozone
potential) specifically for aerosol coatings (California Air Resources Board, 2015b). CARB is also
required to maintain statewide and county/basin-level emissions inventories for 72 source
categories, available 1975 to 2020. Within these categories, the product/process-related sources
include consumer products, architectural coatings, pesticides, cleaning and surface coatings, and
asphalt paving/roofing. Chemically-speciated emissions profiles for sources within these
categories are in the US EPA's SPECIATE repository, some of which are used by the European
Union (EU) (Pernigotti et al., 2016). In the 2015 California code of regulations report, CARB has
further updated regulations focusing on VOC emissions from antiperspirants and deodorants,
hairsprays, other consumer products (both aerosol and non-aerosol), and aerosol coating products
(California Air Resources Board, 2015a).



The Economic Commission of Europe employs market-based mechanisms to reduce regional
emissions of VOCs. The National Emissions Ceilings directive sets country-specific reduction
targets on organic gas emissions ranging from 10% up to 60% from 2010 to 2020 (European
Environment Agency, 2010). The EU has two relevant policy directives: the 'VOC Solvents
Emissions Directive' to limit industrial VOC emissions resulting from processes such as printing,
surface cleaning, vehicular coating, and dry cleaning; and the 'Paints Directive' to reduce VOC
content in paints and varnishes (European Commission, 2014). In China, in addition to restrictions
on criteria pollutants including both $PM_{10}$ and $PM_{2.5}$, the Chinese Air Pollution Control Action
Plan 2013 also limits VOC emissions from paints, adhesives and petrochemical industry, and
promotes the use of low-volatility water-based paints (Ministry of Environmental Protection,

793    2012).


**Appendix B: Calculating characteristic timescales for emissions from surface layers**
The timescale for emission is defined by the simple relation:
$$\tau_{emission} = \frac{M_{applied}}{R_{emission}} \tag{1}$$
The mass applied ($M_{applied}$) and rate of emission ($R_{emission}$) are defined as in Weschler and
Nazaroff (2008), as a function of emission velocity ($v_e$) refers to the airborne mass transfer from
the surface of the applied layer to the free-stream air, which is the rate-limiting step compared to
diffusion within the thin layer (Weschler and Nazaroff, 2008). The gas-phase saturation
concentration immediately above the surface ($C_{sat}$) is determined by molecular structure, $C_O$ is the
concentration of a compound in the layer, A is the exposed surface, and d is the depth of the layer.
$$R_{emission} = v_e A C_{sat} \tag{2}$$





$$M_{applied} = C_O A d \tag{3}$$
Plugging in Eqns. 2-3 into Eqn. 1, and substituting in the partitioning coefficient between octanol
and air ($K_{OA}$), one gets Eqns. 4 and 6: Octanol is chosen as the proxy for the mixed organic layer
that is applied. $K_{OA}$ is available for a wide range of species and is consistent with previous
modeling of SVOC partitioning from surfaces (Weschler and Nazaroff, 2008).
$$\tau_{emission} = \frac{C_O A d}{v_e A C_{sat}} = \frac{C_O d}{v_e C_{sat}} \tag{4}$$
since: $$K_{OA} = \frac{C_o}{C_{sat}} \tag{5}$$
$$\tau_{emission} = \frac{K_{OA} d}{v_e} \tag{6}$$
$v_e$ is a function of aerodynamic and boundary layer resistances ($r_a$ and $r_b$), and can vary
dramatically across indoor and outdoor environments. Given the diversity of compositions,
applications, and environments that products and materials will be applied, constraining $v_e$ is
uncertain and highly variable. Calculations in Table S9 and Figure 4 are shown for a range of $v_e$
from 10 m hr$^{-1}$ to 50 m hr$^{-1}$, which covers a mix of moderately stable to neutral meteorological
conditions. Higher values regularly occur for transport from in/out of some plant canopies. Urban
values are strongly dependent on the urban landscape and regional meteorology; vertical transport
velocities in built up urban areas like Paris range 5-20 m hr$^{-1}$ (Cherin et al., 2015). Indoor values
can decrease to < 5 m hr$^{-1}$ due to lower friction velocities, which is still fast enough to for the
emissions and chemistry that drive indoor air quality (Weschler and Nazaroff, 2008). 10 - 50 m hr$^{-}$
$^{1}$ was chosen as a daytime range, but we acknowledge that slower vertical velocities and thus
longer persistence on surfaces (i.e. lifetimes) may exist for some locations, especially indoors.



Equations 7-11 to calculate $v_e$, $r_a$, and $r_b$ are reproduced below for location/condition-specific
analyses.
$v_e = \frac{1}{r_a + r_b}$ (7)
$r_a = \frac{1}{\kappa u^*} \ln\left(\frac{z_r}{z_0}\right)$ (8)
$r_b = \frac{5 Sc^{2/3}}{u^*}$ (9)
$Sc = v/\mathcal{D}$ (10)
$u^* = \kappa U_r \left[\ln\left(\frac{z_r}{z_0}\right)\right]^{-1}$ (11)
where u*: friction velocity, $z_0$: roughness length, $z_r$: reference height, $U_r$: reference velocity, $\kappa$:
von Karman constant, Sc: Schmidt number, $v$: kinematic viscosity, $\mathcal{D}$: gas diffusivity. Depending
on the relative impact of $r_a$ vs. $r_b$, decreases in diffusivity with larger molecules may affect
transport.





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





**Figure captions**

**Figure 1.** Characteristic evolution and modernization of in-use motor vehicle fleets. Trends in motor vehicle

population, fuel use, and VOC emissions for model years conforming to pre-LEV, LEV-I, -II, and -III emissions

standards in California's South Coast air basin for 1990-2020 from CARB's EMFAC database (California Air

Resources Board, 2014).

**Figure 2.** Emissions and potential air quality impacts of product/process-related sources and motor vehicles over

time in the South Coast air basin (i.e. Los Angeles). (a) Total VOC emissions (excluding ethane). (b) Potential SOA

from product/process-related emissions compared to on-road motor vehicles, with uncertainties based on the ranges

of compound class-specific SOA yields, and (c) Ozone formation potential via compound-specific maximum ozone

incremental reactivity values (SAPRC, no uncertainty given). Product/process-related results are produced using

CARB emissions data and EPA source profiles; see methods for further detail. Note: None of the panels include

VOCs, IVOCs, or SVOCs, from products/processes that we identify in Sections 2 and 4 as missing from emissions

inventories, but SOA and ozone formation from motor vehicle emissions does include I/SVOCs. Maximum potential

SOA from gasoline vehicles in 1990 was ~41 tons day$^{-1}$ (off graph). Pre-1990 potential SOA and ozone for motor

vehicles is excluded due to the lack of fleet-resolved data.

**Figure 3.** Average emissions from product/process-related sources for 2005-2020 based on data from CARB

inventory and SPECIATE database as a function of (a) compound class (shown with standard deviations) and (b)

major product/process-related source categories in the inventory (values in Table S3). "Miscellaneous" contains

70% mineral spirits ($C_{7-12}$ hydrocarbons). I/SVOCs does not include an estimate of the missing emissions identified

in Section 2 and 4.

**Figure 4.** Emission timescales for single-ring aromatics, alkanes, and prominent solvents: ethanol (a), ethylene

glycol (b), and acetone (c). Based on applied layers of 0.01-1 mm and vertical transport coefficients for

neutral/stable outdoor conditions (10-50 m hr$^{-1}$), with longer timescales in indoor environments or thicker layers



(e.g. asphalt, building materials). The left axis is reproduced on the right in days. Timescales are limited by gas-
phase transport from the surface, but absorption into polymeric or porous substrates could extend timescales
(Weschler and Nazaroff, 2008). Volatility range boundaries for VOC-IVOC and IVOC-SVOC occur between $C_{12}$-
$C_{13}$ and $C_{19}$-$C_{20}$, respectively, for n-alkanes or compounds with equivalent volatilities. See Appendix B for
calculations and Table S9.

**Figure 5.** Comparison of (a) SOA yields and (b) Ozone formation potential of five major sources. Blue markers
represent average yields of the product/process-related source categories in Figure 3, and do not include "missing"
emissions. The red markers show yields of sampled consumer products calculated from speciation obtained after
GC-MS analysis. All are shown as a function of mass emitted, not product composition.





**FIGURES & TABLES**
**Table 1.** Composition and emittable fraction of twelve commercially available consumer products, and results of
their carbon isotope analysis. Please see Table S1 for detailed mass distribution profiles and Table S7 for detailed
carbon isotopic analysis results.

| Product name | Percent Fossil Origin | IVOC Content | Aromatic Content | Emittable Fraction | d¹³C (‰) | Δ¹⁴C (‰) |
|---|---|---|---|---|---|---|
| Naphtha cleaner | >99% | - | - | 100% | -29.0 ± 0.1 | -999.4 ± 0.5 |
| Non-polar solvent | >99% | - | 93% | 100% | -27.4 ± 0.1 | -998.5 ± 0.5 |
| Fogging Insecticide[†] | >99% | 95% | - | 95% | -26.7 ± 0.1 | -999.1 ± 0.5 |
| Semi-gloss furniture coating | 56% | - | 6% | 30% | -29.9 ± 0.1 | -565.4 ± 0.7 |
| Multipurpose solvent[a] | >99% | - | - | 100% | -30.6 ± 0.1 | -997.8 ± 0.7 |
| Furniture coating[†] | 65% | - | 1% | 40% | -30.2 ± 0.1 | -650.0 ± 0.6 |
| Roof paint[b] | >98% | - | - | - | -22.7± 0.1 | -985.0 ± 0.5 |
| Sealant[c, §] | 97% | 0.77% | 21% | 25% | -27.4 ± 0.1 | -969.9 ± 0.5 |
| Paint thinner[†] | >99% | 2% | 3% | 100% | -29.1 ± 0.1 | -999.7 ± 0.5 |
| Asphalt coating[d, §] | >99% | 4% | 3% | 25% | -27.6 ± 0.1 | -994.8 ± 0.6 |
| Detergent[e] | 81% | 3.5% | 4% | 25% | -27.9 ± 0.1 | -807.8 ± 0.6 |
| General purpose cleaner[f, †] | 81% | 1% | - | 3% | -27.7 ± 0.1 | -811.1 ± 1.3 |
| Multipurpose lubricant | - | 39% | - | 97% | - | - |
| Aerosol Coating Product[g, †] | - | - | 12% | 17% | - | - |
| Flashing cement[h, §] | - | 0.2% | 3.5% | 27% | - | - |
| Crawling Insecticide 1[i, †] | - | 20% | - | 20% | - | - |
| Crawling Insecticide 2[j, †] | - | 8% | - | 8% | - | - |

*Crude oil and plant-derived ethanol were used for multiple reference blanks (Table S8).
§Asphalt-related products.
†Applied as aerosols.
[a] Largely consists of acetone (80%) with the rest as cyclotetrasiloxanes and aryl halides.
[b] Water based product. Other components include titanium dioxide, silica and aluminum hydroxide.
[c] Largely consists of petroleum asphalt, clays and cellulose.
[d] Largely consists of petroleum asphalt.
[e] 46% of the emittable fraction (EF) consists of mostly esters, 70% of which have less than 12 carbon atoms. Terpenes including limonene, eucalyptol and α-terpineol form 37% of the EF.
[f] Water-based product. Also has terpenes including camphene, d3-carene, α-pinene, linalool and δ-limonene. 60% of terpene fraction is δ-limonene.
[g] Contains 35% acetone and 25% hydrocarbon propellants not included in the EF. *25% of EF is 75% acetate + 25% ketones.
[h] Largely consists of petroleum asphalt, kaolin, cellulose and aluminum magnesium silicate.
[i] Contains 15% hydrocarbon propellants not included in the EF.
[j] Contains 25% hydrocarbon propellants not include in the EF.
Note: Emittable fraction is confirmed with MSDS where possible.






**Table 2.** Analysis of material safety data sheets commercially available with products.

| Product Category | Fraction of products with aromatic content | MSDS Aromatics %wt. Mean (range) | Fraction of products with I/SVOC content | MSDS I/SVOCs %wt. Mean (range) |
|---|---|---|---|---|
| Paints* | 0% | 0 | 24%[a] | 2.5 (1-5) |
| Adhesives | 27% | 8 (1-30) | 7%[b] | 20 (10-30) |
| Cleaning Products | 8% | 5 (1-10) | 23%[c] | 3.5 (1-7) |
| Sealants | 53% | 9 (1-30) | 7%[d] | 4 (1-10) |
| Pesticides** | NS | NS | NS | NS |

[a]Consists of benzoates. Log $K_{OA}$=7.75 which puts them in the IVOCs range.
[b]Does not include potential emissions from petroleum asphalt which on average constitutes 36% in 13% of the surveyed products. 50% products contain 38% limestone on average.
[c]Includes C9-C14 ethoxylated alcohols.
[d]Does not include potential emissions from petroleum asphalt which on average constitutes 40% in 33% of the surveyed products. 60% products contain 36% limestone on average.
*20% of MSDSs include 'non-hazardous' or 'proprietary' component in composition ranging from 60%-100%.
**MSDSs do not present sufficient information on the composition of pesticides (NS: not sufficient).




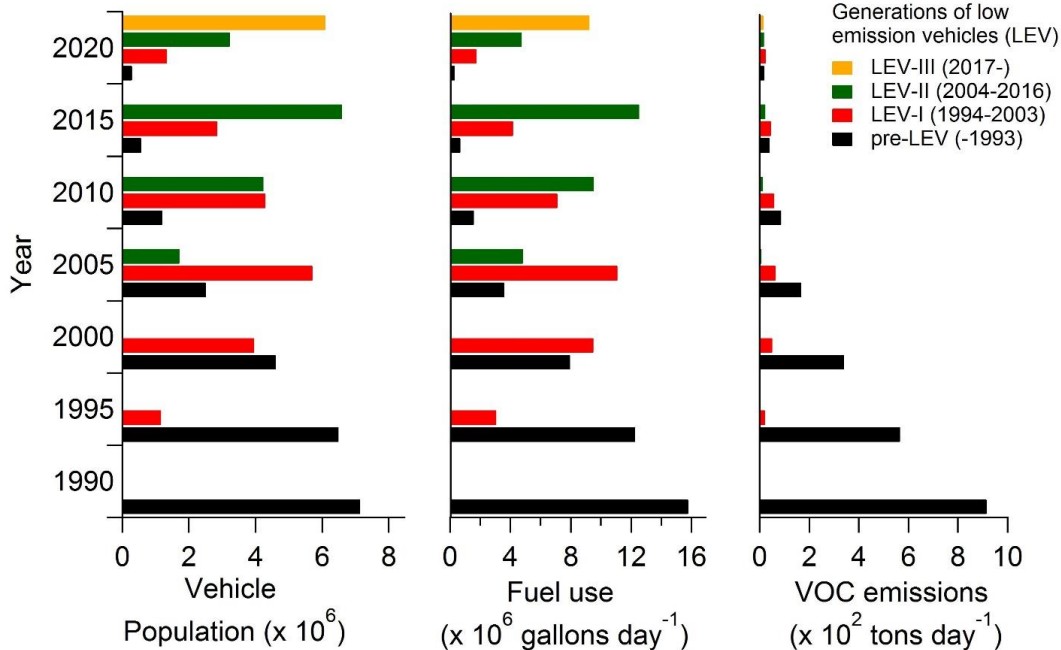



**Figure 1.** Characteristic evolution and modernization of in-use motor vehicle fleets. Trends in motor vehicle

population, fuel use, and VOC emissions for model years conforming to pre-LEV, LEV-I, -II, and -III emissions

standards in California's South Coast air basin for 1990-2020 from CARB's EMFAC database (California Air

Resources Board, 2014).





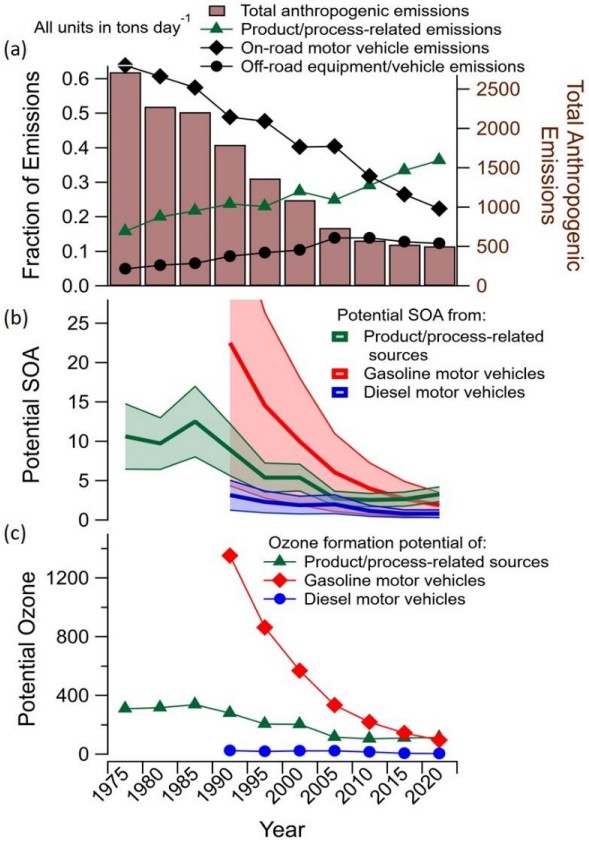

**Figure 2.** Emissions and potential air quality impacts of product/process-related sources and motor vehicles over

time in the South Coast air basin (i.e. Los Angeles). (a) Total VOC emissions (excluding ethane). (b) Potential SOA

from product/process-related emissions compared to on-road motor vehicles, with uncertainties based on the ranges

of compound class-specific SOA yields, and (c) Ozone formation potential via compound-specific maximum ozone

incremental reactivity values (SAPRC, no uncertainty given). Product/process-related results are produced using

CARB emissions data and EPA source profiles; see methods for further detail. Note: None of the panels include

VOCs, IVOCs, or SVOCs, from products/processes that we identify in Sections 2 and 4 as missing from emissions

inventories, but SOA and ozone formation from motor vehicle emissions does include I/SVOCs. Maximum potential

SOA from gasoline vehicles in 1990 was ~41 tons day$^{-1}$ (off graph). Pre-1990 potential SOA and ozone for motor

vehicles is excluded due to the lack of fleet-resolved data.

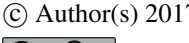



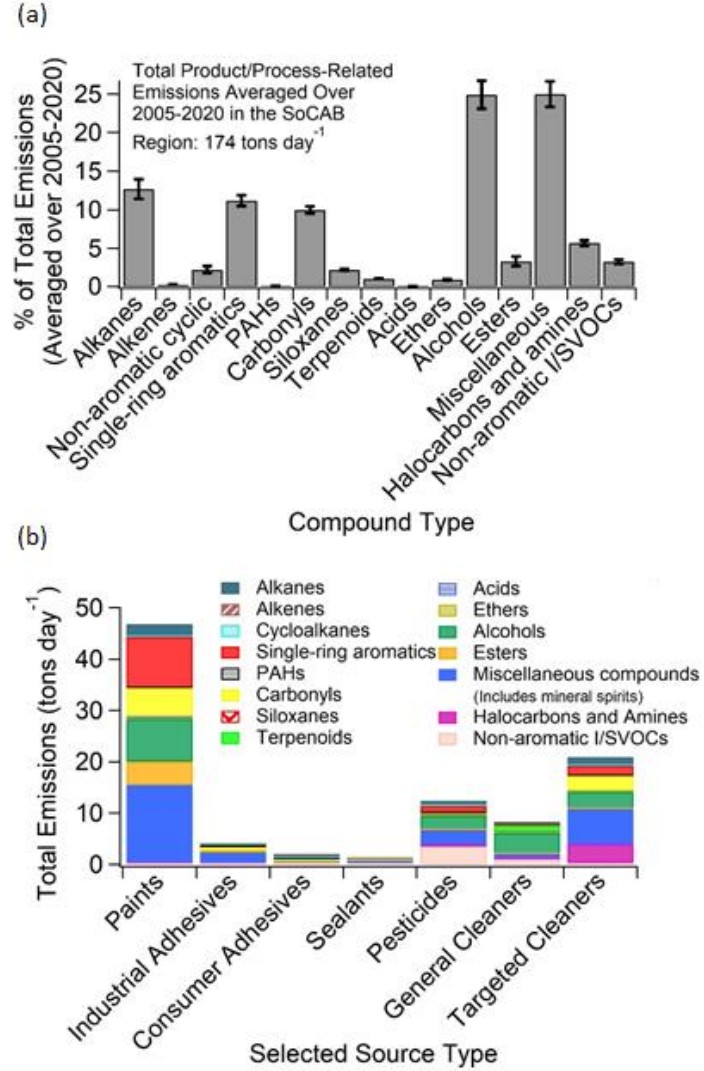



**Figure 3.** Average emissions from product/process-related sources for 2005-2020 based on data from CARB
inventory and SPECIATE database as a function of (a) compound class (shown with standard deviations) and (b)
major product/process-related source categories in the inventory (values in Table S3). "Miscellaneous" contains
70% mineral spirits ($C_{7-12}$ hydrocarbons). I/SVOCs does not include an estimate of the missing emissions identified
in Section 2 and 4.





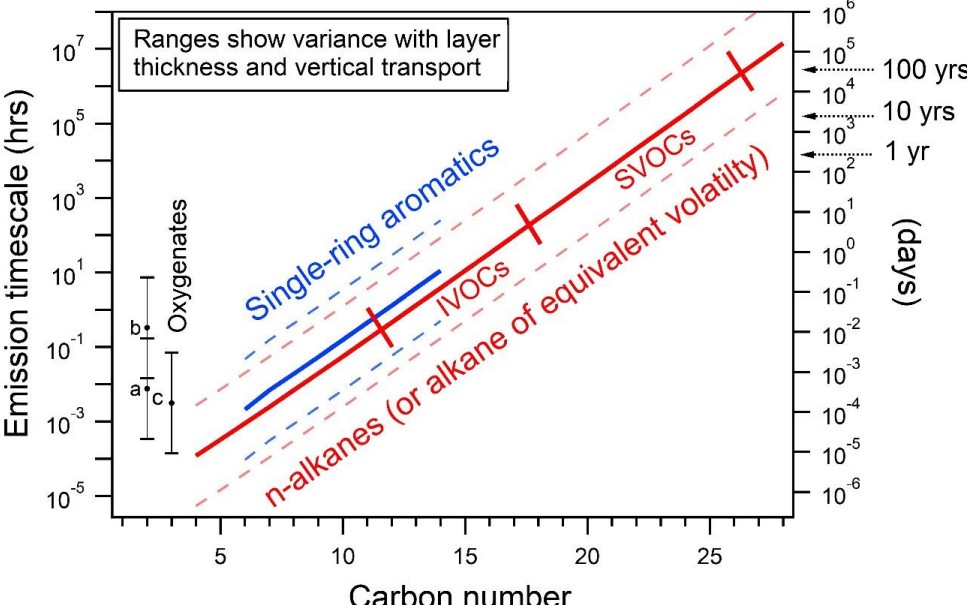


**Figure 4.** Emission timescales for single-ring aromatics, alkanes, and prominent solvents: ethanol (a), ethylene

glycol (b), and acetone (c). Based on applied layers of 0.01-1 mm and vertical transport coefficients for

neutral/stable outdoor conditions (10-50 m hr$^{-1}$), with longer timescales in indoor environments or thicker layers

(e.g. asphalt, building materials). The left axis is reproduced on the right in days. Timescales are limited by gas-

phase transport from the surface, but absorption into polymeric or porous substrates could extend timescales

(Weschler and Nazaroff, 2008). Volatility range boundaries for VOC-IVOC and IVOC-SVOC occur between $C_{12}$-

$C_{13}$ and $C_{19}$-$C_{20}$, respectively, for n-alkanes or compounds with equivalent volatilities. See Appendix B for

calculations and Table S9.

1392



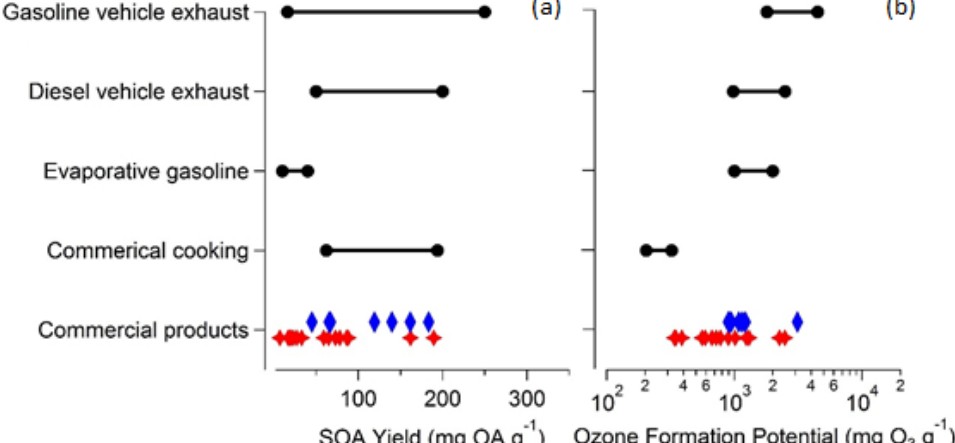

1393

1394

**Figure 5.** Comparison of (a) SOA yields and (b) Ozone formation potential of five major sources. Blue markers

represent average yields of the product/process-related source categories in Figure 3, and do not include "missing"

emissions. The red markers show yields of sampled consumer products calculated from speciation obtained after

GC-MS analysis. All are shown as a function of mass emitted, not product composition.

1399

1400