# Peer review of "Considering the future of anthropogenic gas-phase organic compound"

_Atmospheric Chemistry and Physics, 2017_

## Referee Comment (RC1) · Anonymous Referee #1 · 15 Feb 2018

Review 'Considering the future of anthropogenic gas-phase organic compound emissions and the increasing influence of non-combustion sources on urban air quality'

ACP-2017-761

This paper presents a detailed look at VOCs, specifically delving into their representation in emission inventories, what details are (or are not reported) relating to composition, and their ozone and SOA formation potential. The authors spend some time as well to investigate intermediate-volatility and semivolatile organic compounds that are typically not included in standard emission inventories and are exempt from emissions targets, but in need of greater attention in the future. Finally, the misattribution of some of the VOCs to fossil fuel from e.g., vehicles, while some likely originates from e.g., petrochemical processes, which also have a fossil fuel feedstock, needs to be considered in source attribution.

This paper adds interesting and novel points to the literature and is generally well written and well organized, if a bit long and drawn out at points. I would recommend publication after addressing some minor points for revision.

Comments:

L116-118: the use of the 'pp' prefix seems unnecessary given that it is then only used ca. 20 pages later and only twice. When it is initially explained here the point seems odd and unnecessary. When it is then applied much later, as a reader you have to go back and figure out where that came from so that you are clear what is meant. It would be clearer to just remove the initial text and just explain the 'products and processes' source when discussed at that one point later in the text.

Overall, the paper has both appendixes and SI. It is not clear what the criteria are for having information in an appendix versus in the SI. I would suggest to combine these all into the SI. Furthermore, there are a number of figures and tables in the SI that are referred to quite often in the text. I would suggest that the authors consider moving some of these to the main text. Supplemental information should really only be for information that provides a level of detail that most readers will not be interested in, but is important for those who e.g., might want to apply some of the methods to their own analysis or really dig into the details. For example, I would suggest moving e.g., Table S2 or Figure S3 to the main text. If there is a concern that there are too many figures/tables in the main text, you could move figure 4 to the SI, which I find adds less than some of the other figures.

L245-246: the SOA yield estimate – what is this value based on? And is this in general or specifically for the unstudied compounds? More information would be good here to clarify what is meant and where these numbers come from.

L248-250: can you give some information as to the magnitude of this? Are we talking an order of magnitude or 10%? A rough indication is fine.

L269-270: please add some more information to clarify how the composition is calculated. The source categories are based on the CARB inventory, but where is the speciation coming from? Or are you explaining what you will be doing with the speciate database and then MSDSs etc that is explored in the following sections. This isn't clear.

L313-315: Can you provide some more information as to the comparison to the speciate profiles? Percent of single-ring aromatic content is listed in the text for a couple of products and for more in Table 2, but a comparison percent is not provided for the speciate source profiles. There is a comparison listed, but this seems to be from a different source and not speciate. If this is the comparison to speciate, please be explicit about this. One could also consider adding information to the table.

L315, but also more generally (L337, L342, L371, etc): sometimes the term solvents is used, at other times consumer products, sometimes product/process-related VOCs. How these terms are used and where the overlap is or is not, needs to be defined. This would really help the clarity of the manuscript. In some cases (not necessarily how it is used here), solvents is an umbrella classification for consumer products, paints, etc. But it could also be a more limited definition.

L390: is there any indication of how much is missing. You cite how much is included, but can the missing amount be estimated at all from your analysis? Even if it is only for a couple of products or is just an order of magnitude range?

L509/Figure 5: as stated in the text and in the figure caption, the product emissions have SOA yields and ozone formation potentials on par with other major urban sources as a function of mass emitted. What about overall. It would be good to explicitly include this information based on your EI estimates here, also with a comment on whether you find the sources to be underestimated or not, as these are really some of the important implications of this work.

The last two italicized sub-sections in section 4.2 seem a bit disjointed from the rest. With the first one on off-road combustion, I see what you are getting at with this, to just show the importance of off-road in the broader context, also for products/processes, but it is quite long and drawn out. I think this could be shortened significantly and the same points made (also because this really isn't the main aim of the paper at all). Following on from that the section on modifying factors then seems pretty random and short. Could these points be integrated elsewhere? Possibly in the beginning of 4.2 as part of an intro/context for SOA/ozone formation?

L671/Inclusion of IVCOs… section: This section could be shortened a bit and made more concise.

Edits:

L144: evaporation of a solvent from a product or during a process – this is a bit awkward. From a product, is it meant here during storage? Or during use? Can this be clarified?

L168: why is 'curing' in quotes and none of the others are?

L196-197: 'other additional methods details' such as what? It would be good to mention what those are that are relevant to the text just outlined.

L230: '….the U.S. National Emissions Inventory and emission inventories from the Global Emissions Initiative (GEIA) were used for….'

L236: write out EMFAC

L237: write out LEV

L275: '…such as those discussed in section 4.' We are in section 4 and it has loads of subsections and examples, etc. so can you be more specific?

L281: here, as well as some other cases (e.g., L450), please add a 'from' before listing a % or concentration range. E.g., '… from 3-100% …'

L308: reference Fig 3b as well as S4 because they essentially show the same thing. The only thing that S4 adds is the uncertainty(?) or standard deviation(?) associated with the compound classes. Please specify in the figure caption for S4 what the error bars indicate.

L318: add '…in the U.S.' at the end of the sentence unless this is inappropriate, in which case please specify the region.

L326: '….where provided' (currently just provide)

L426: 18-29 Gg, of how many total Gg VOC emissions per year in CA? Is this a lot or not? Context would be good to add here (even if the info is available elsewhere).

L442: larger asphalts? What is meant by this? Larger VOCs from asphalts that degrade to form smaller VOCs?

L459-460: please define 'cutback asphalt'

L461-463: how are these references also relevant for the EU inventories?

L516: is the additional 25% reduction relative to 1990, or relative to the 2015 amount?

L535: only 5% for which year?

L605: 'their emissions' can you please be explicit and mention which ones? it is not clear here.

L621: it would be good to give an e.g., and mention some of the other anthropogenic sources explicitly.

L636-637: This sentence is awkward and needs to be rewritten.

L663: might be good to reiterate the three pathways from section 2.

---

## Author Comment (AC1) · 27 Feb 2018

We sincerely thank the reviewer and editor for supporting our manuscript, and insights which have helped us in improving our work. After having gone through the comments in great detail, we are writing to present our revisions/replies. The reviewer comments and our responses are presented below.

Reviewer 1:

This paper presents a detailed look at VOCs, specifically delving into their represen-

tation in emission inventories, what details are (or are not reported) relating to composition, and their ozone and SOA formation potential. The authors spend some time as well to investigate intermediate-volatility and semivolatile organic compounds that are typically not included in standard emission inventories and are exempt from emissions targets, but in need of greater attention in the future. Finally, the misattribution of some of the VOCs to fossil fuel from e.g., vehicles, while some likely originates from e.g., petrochemical processes, which also have a fossil fuel feedstock, needs to be considered in source attribution.

This paper adds interesting and novel points to the literature and is generally well written and well organized, if a bit long and drawn out at points. I would recommend publication after addressing some minor points for revision.

Comments:

-L116-118: the use of the 'pp' prefix seems unnecessary given that it is then only used ca. 20 pages later and only twice. When it is initially explained here the point seems odd and unnecessary. When it is then applied much later, as a reader you have to go back and figure out where that came from so that you are clear what is meant. It would be clearer to just remove the initial text and just explain the 'products and processes' source when discussed at that one point later in the text.

Response: We agree that the prefix 'pp' could be dispensed with given its minimal use in the manuscript. It is now completely removed from the revised manuscript. In places where it previously appeared, 'pp' is replaced with the term 'products/process-related' which has been frequently used in the manuscript.

-Overall, the paper has both appendixes and SI. It is not clear what the criteria are for having information in an appendix versus in the SI. I would suggest to combine these all into the SI.

Response: We understand the reviewer's concern and reviewed the ACP author guide-

lines on the matter. Appendices A and B in our manuscript provide valuable and directly relevant information to the average reader regarding the historical context and significance of product/process-related emissions over long timescales, but don't seem to qualify as SI based on the author guidelines. They are shown in the appendices rather than the main text to help maintain a reasonable length for the main body of the manuscript. Hence, we prefer to keep the two appendices with the published manuscript while carefully complying with the ACP author guidelines. We defer to the editor's judgement on this matter.

-Furthermore, there are a number of figures and tables in the SI that are referred to quite often in the text. I would suggest that the authors consider moving some of these to the main text. Supplemental information should really only be for information that provides a level of detail that most readers will not be interested in, but is important for those who e.g., might want to apply some of the methods to their own analysis or really dig into the details. For example, I would suggest moving e.g., Table S2 or Figure S3 to the main text. If there is a concern that there are too many figures/tables in the main text, you could move figure 4 to the SI, which I find adds less than some of the other figures.

Response: We agree with the reviewer that any SI figures and tables that are referenced frequently in the main text should appear in the main text, but we reviewed the manuscript's references to SI figures/tables, and we wish to highlight that tables and figures listed in the SI are individually cited once or twice at most in the main manuscript. The emission timescales shown in Figure 4 are very important for expanding understanding of the timescales of emissions and long-term I/SVOC emissions from products and processes, and their lack of consideration in existing inventories/models, which is part of the core theme of our manuscript. Additionally, Figure 4 is referenced 5 times in the manuscript. Therefore, considering these reasons and reviewer's concern regarding the manuscript length, we have decided to keep the figures and tables as it is.

-L245-246: the SOA yield estimate – what is this value based on? And is this in general or specifically for the unstudied compounds? More information would be good here to clarify what is meant and where these numbers come from.

Response: We understand that this is an important concern. We have clarified this point in the text and directed the reader to Table S6 where the rules that we used for assigning the SOA yield values are listed in great detail. These yield values are applicable to unstudied organic compounds and are based on trends observed in published yields of such organic compounds that are comparable in terms of molecular structure. In cases of compounds where exact SOA yields are straight up available from scientific literature, they are directly used.

-L248-250: can you give some information as to the magnitude of this? Are we talking an order of magnitude or 10%? A rough indication is fine.

Response: Unfortunately, it is difficult to generalize and constrain based on the current state of knowledge, especially relevant to urban air quality. There are too many different precursors to estimate since the impact is likely to be precursor-dependent and site-dependent.

-L269-270: Please add some more information to clarify how the composition is calculated. The source categories are based on the CARB inventory, but where is the speciation coming from? Or are you explaining what you will be doing with the speciate database and then MSDSs etc that is explored in the following sections. This isn't clear.

Response: We thank the reviewer for pointing this out. We include a more detailed explanation in lines 223-226 of the (revised) manuscript, we combine source categories from the CARB Almanac emissions inventory with the chemical profiles for those sources listed in the EPA SPECIATE 4.4 database to calculate compound-specific emissions in the south coast air basin (SoCAB). We have also added an SI figure (figure S5) for comparison using an alternative speciation profile (applicable to limited

years), with very similar results. The MSDSs were not used to establish source profiles. We have made revisions to clarify the primary data source in the section of concern.

-L313-315: Can you provide some more information as to the comparison to the speciate profiles? Percent of single-ring aromatic content is listed in the text for a couple of products and for more in Table 2, but a comparison percent is not provided for the speciate source profiles. There is a comparison listed, but this seems to be from a different source and not speciate. If this is the comparison to speciate, please be explicit about this. One could also consider adding information to the table.

Response: In lines 310-311, we state that ∼21% of emissions from paints are single-ring aromatics. However, we understand the need for more clarity and information in the sentence in question. Figure 3(b) and figure S4 are now cited in line 317 of the revised manuscript to point the reader to relevant SPECIATE data to help compare with our findings of the MSDS surveys.

-L315, but also more generally (L337, L342, L371 etc.): sometimes the term solvents is used, at other times consumer products, sometimes product/process-related VOCs. How these terms are used and where the overlap is or is not, needs to be defined. This would really help the clarity of the manuscript. In some cases (not necessarily how it is used here), solvents is an umbrella classification for consumer products, paints, etc. But it could also be a more limited definition.

Response: We agree with the reviewer that the mix of terms used by the field has been confusing and sometimes misleading. We use the products/processes as the general inclusive term for this diverse range of sources. However, historically, research and reporting has been focused narrowly on either just "solvents" or "consumer products". At the lines in question, we purposefully report prior work and data with the classification they used for the sake of consistency and accuracy. We have tried to improve the wording in line 318 to prevent confusion.

-L390: is there any indication of how much is missing. You cite how much is included,

but can the missing amount be estimated at all from your analysis? Even if it is only for a couple of products or is just an order of magnitude range?

Response: In an attempt to constrain the potential IVOC content of current products, we provide this information in the 'IVOC content' column in tables 1 and 2 via different approaches, and discuss it later in the manuscript.

-L509/Figure 5: as stated in the text and in the figure caption, the product emissions have SOA yields and ozone formation potentials on par with other major urban sources as a function of mass emitted. What about overall. It would be good to explicitly include this information based on your EI estimates here, also with a comment on whether you find the sources to be underestimated or not, as these are really some of the important implications of this work.

Response: We agree that this would also provide useful information. Yet, we present this figure as shown/described since it is conventional to report SOA yields as a function of mass of precursor species reacted, which is directly connected to mass emitted, and less to total mixture mass of product used/applied. The emissions inventories also only report mass emitted and not mass used/applied so it is not possible to back-calculate the requested information from the emissions inventory analysis. As to whether or not these yields are underestimated relative to potential emissions, direct emissions and their potential SOA via pathways #1 and #2 (i.e. solvents + solute evaporation) is included, but emissions and potential SOA from degradation byproducts (pathway #3) are not since the larger precursors to degradation b-products have little- to no-volatility and were not eluted from the column. So in that the respect, these are conservative estimates. We acknowledge this pathway inclusion/exclusion in the paper.

-The last two italicized sub-sections in section 4.2 seem a bit disjointed from the rest. With the first one on off-road combustion, I see what you are getting at with this, to just show the importance of off-road in the broader context, also for products/processes,

but it is quite long and drawn out. I think this could be shortened significantly and the same points made (also because this really isn't the main aim of the paper at all). Following on from that the section on modifying factors then seems pretty random and short. Could these points be integrated elsewhere? Possibly in the beginning of 4.2 as part of an intro/context for SOA/ozone formation?

Response: We understand the reviewer's comments, and have tried to shorten the section on off-road emission sources. However, we feel that these two subsections are key points of discussion as they play central roles in emissions and atmospheric chemistry in urban areas worldwide. Off-road mobile sources are important factors in urban air quality that have been mitigated at different timelines to on-road mobile sources. After reviewing and editing the section, we have decided to leave them in place since they are extremely important considerations that future researchers or policy-makers need to take into account when applying the methods or interpreting our findings in the context of specific urban areas outside the scope of our study.

-L671/Inclusion of IVCOs. . . section: This section could be shortened a bit and made more concise.

Response: At the reviewer's request, we have revised and shortened this section in lines 689-713 of the (revised) manuscript.

Edits:

-L144: evaporation of a solvent from a product or during a process – this is a bit awkward. From a product, is it meant here during storage? Or during use? Can this be clarified?

Response: Done. It is rephrased in line 143 of the revised manuscript for clarity.

-L168: why is 'curing' in quotes and none of the others are?

Response: Edited. Quotes removed in line 168.

-L196-197: 'other additional methods details' such as what? It would be good to mention what those are that are relevant to the text just outlined.

Response: Edited in line 197. Part of the line in question now reads "...and other additional methods details pertaining to potential SOA and ozone estimation can be found in the supporting information."

-L230: '....the U.S. National Emissions Inventory and emission inventories from the Global Emissions Initiative (GEIA) were used for....'

Response: Thank you for pointing this out. A correction has been made in line 231 of the (revised) manuscript.

-L236: write out EMFAC

Response: In lines 237-238, 'EMFAC' has been changed to 'Emissions Factor (EMFAC)'.

-L237: write out LEV

Response: In line 239, 'LEV' has been changed to 'low-emission vehicles (LEV)'.

-L275: '...such as those discussed in section 4.' We are in section 4 and it has loads of subsections and examples, etc. so can you be more specific?

Response: Done. Revised in line 278.

-L281: here, as well as some other cases (e.g., L450), please add a 'from' before listing a % or concentration range. E.g., '... from 3-100% ...'

Response: We thank the reviewer for mentioning this. A correction has been made in line 284 now.

-L308: reference Fig 3b as well as S4 because they essentially show the same thing. The only thing that S4 adds is the uncertainty (?) or standard deviation (?) associated with the compound classes. Please specify in the figure caption for S4 what the error

bars indicate.

Response: In line 311, figure 3(b) is now cited along with figure S4. The figure caption of figure S4 is edited to include that the error bars represent standard deviation in emissions from paints between the years 2000 and 2020.

-L318: add '…in the U.S.' at the end of the sentence unless this is inappropriate, in which case please specify the region.

Response: In line 322, 'in the U.S.' is added at the end of the sentence.

-L326: '….where provided' (currently just provide)

Response: Correction made in line 329.

-L426: 18-29 Gg, of how many total Gg VOC emissions per year in CA? Is this a lot or not? Context would be good to add here (even if the info is available elsewhere).

Response: We have added more information in lines 436-439 to show the importance of pesticide emissions as a source of air pollution.

-L442: larger asphalts? What is meant by this? Larger VOCs from asphalts that degrade to form smaller VOCs?

Response: In this sentence, we indeed point toward the non-solvent emissions that are caused by the degradation (i.e. fragmentation) of larger organic compounds from asphalts to form smaller compounds. The sentence has been edited in line 455 of the revised manuscript.

-L459-460: please define 'cutback asphalt'

Response: Cutback asphalt is now defined in line 473 of the revised manuscript.

-L461-463: how are these references also relevant for the EU inventories?

Response: We thank the reviewer for pointing this out. We have now added a reference relevant to the EU inventories in line 475 of the revised manuscript.

-L516: is the additional 25% reduction relative to 1990, or relative to the 2015 amount?

Response: As stated in line 530, by 2020, a 25% reduction in potential SOA from on-road gasoline vehicles is expected relative to the year 2015's value of 3.3 tons day-1.

-L535: only 5% for which year?

Response: This is true for the year 2015. The sentence is now edited for clarity in line 550.

-L605: 'their emissions' can you please be explicit and mention which ones? it is not clear here.

Response: In line 620, 'their emissions' is replaced with 'products/process-related emissions' for clarity.

-L621: it would be good to give an e.g., and mention some of the other anthropogenic sources explicitly.

Response: We have revised lines 634-638 of the (revised) manuscript to clarify the conclusions of the cited work.

-L636-637: This sentence is awkward and needs to be rewritten.

Response: Done. The sentence is now revised in lines 652-654.

-L663: might be good to reiterate the three pathways from section 2

Response: Done. The sentence is rewritten in lines 679-681 to reiterate the three emission pathways.

---

## Referee Comment (RC2) · Anonymous Referee #2 · 28 Feb 2018

Following on from the detailed reviewer comments added on the 15th of February, I have only a few things to add. I think this paper is a valuable contribution to the literature and nicely complements the paper by McDonald et al. recently published in Science (I was surprised there was no reference to this paper here).

Line 124: Should furnishings be added to your list? There are emissions of VOCs from e.g carpets that I'm not entirely clear are included in your list. Is it also worth somewhere making the distinction between primary and secondary emissions from surface sources, as these will vary over time – so primary emissions from furnishings/building

materials are high in the first instance and then decrease over time, whilst secondary emissions are ozone dependent and may even increase over time if outdoor ozone concentrations (and hence indoor concentrations) increase.

Line 261: The products you investigated have 'hidden' ingredients (varying from 30-60% of the total). Is it similar for similar types of products, so you could potentially estimate the amount that is missing from your inventory, or is it manufacturer rather than product dependent or random ?

Line 498: There is some literature that suggests siloxanes may have health impacts - see review of such by Tran and Kannan in Science of the Total Environment 511 (2015) 138–144.

Section 4.2: I agree with the first reviewer that this section is too long. I wasn't clear from your response whether you had shortened this section significantly, but if not, would urge you to make it shorter and more focused.

Table 1: I was a bit confused by some of the footnotes. You need to tie them more specifically to the information in the table so we know which fraction the footnotes refer to (e.g. footnote f refers to the emitted fraction presumably?).

The authors might also want to consider the increasing use of so-called 'green' materials indoors that typically have lower emission rates than more traditional materials. If such materials become more widely adopted, it may be that indoor sources of VOCs decrease over the considered period of time as well (or at least those from some of the sources considered).

Edits

Line 322: Sentence too long and grammar needs improving.

Line 703: grammar needs improving.

---

## Author Comment (AC2) · 1 Mar 2018

We thank the reviewer for supporting our manuscript and their insightful comments which have helped us further improve our work. After having gone through the comments in detail, we present our revisions/replies below:

Reviewer 2:

Following on from the detailed reviewer comments added on the 15th of February, I have only a few things to add. I think this paper is a valuable contribution to the

literature and nicely complements the paper by McDonald et al. recently published in Science (I was surprised there was no reference to this paper here).

-Line 124: Should furnishings be added to your list? There are emissions of VOCs from e.g carpets that I'm not entirely clear are included in your list. Is it also worth somewhere making the distinction between primary and secondary emissions from surface sources, as these will vary over time – so primary emissions from furnishings/building materials are high in the first instance and then decrease over time, whilst secondary emissions are ozone dependent and may even increase over time if outdoor ozone concentrations (and hence indoor concentrations) increase.

Response: We agree with the reviewer that furnishings are a major potential source of emissions that belong as part of this work. We had broadly included these things with building materials but acknowledge that greater detail could be provided. To address this concern, we have modified the bullet points in lines 131-132 to "Building materials (e.g. carpeting, flooring, insulation, wood, gypsum)" and "Furnishings (e.g. furniture)". Secondly we also agree that primary versus secondary emissions are an important differentiation. We include them both in our three emission pathways described in section 2, lines 144-149. To address the reviewer's concern, we have changed emission pathway #3 in lines 148-149 to also include emissions via degradation of "a solid product/material". We also refer the author to lines 156-159 where we specifically discuss emissions via ozone oxidation, and have now added a comment about emission timescales for these emissions.

-Line 261: The products you investigated have 'hidden' ingredients (varying from 30-60% of the total). Is it similar for similar types of products, so you could potentially estimate the amount that is missing from your inventory, or is it manufacturer rather than product dependent or random?

Response: Our survey of MSDSs demonstrated very wide ranges of reported product composition for proprietary reasons. The reported ranges tended to vary at random,

and could not be reliably constrained by product type.

-Line 498: There is some literature that suggests siloxanes may have health impacts - see review of such by Tran and Kannan in Science of the Total Environment 511 (2015) 138–144.

Response: We thank the reviewer for pointing out the Tran and Kannan (2015) paper. We acknowledge potential health effects from a wide range of compound classes that might not be fully understood. We intended "low" to include things that are not part of well-established hazardous compounds such as those on the air toxics list. We have now edited the sentence in lines 514-516 of the (revised) manuscript for clarity, and removed "siloxanes" to resolve the reviewer's concern.

-Section 4.2: I agree with the first reviewer that this section is too long. I wasn't clear from your response whether you had shortened this section significantly, but if not, would urge you to make it shorter and more focused.

Response: We have revisited the last 2 sub-sections of section 4.2 again, and based on the comments from the two reviewers, we have significantly shortened the "Contributions from off-road combustion-related sources" sub-section to a single, more concise paragraph. We have also incorporated the sub-section on modifying factors into the introduction (as suggested by Reviewer #1).

-Table 1: I was a bit confused by some of the footnotes. You need to tie them more specifically to the information in the table so we know which fraction the footnotes refer to (e.g. footnote f refers to the emitted fraction presumably?).

Response: We agree with the reviewer and have rephrased the "Note" at the bottom of Table 1 footnotes to prevent confusion. The "Note" now reads, "All fractions given in the footnotes refer to the emittable fraction of a product. Emittable fraction is confirmed with MSDS where possible."

-The authors might also want to consider the increasing use of so-called 'green' materials indoors that typically have lower emission rates than more traditional materials. If such materials become more widely adopted, it may be that indoor sources of VOCs decrease over the considered period of time as well (or at least those from some of the sources considered).

Response: We agree with the reviewer that this is a very interesting area of development within the market and research field, but it is unclear if a holistic consideration of emissions across all three described pathways will yield significantly different emissions/conclusions, since modifications have primarily targeted more direct emissions historically. For example, a study by Toftum et al. 2008 (cited in the manuscript in line 161) reports SOA formation from ozonation of a green paint. We agree that this is a very interesting area of research and have acknowledged it in the conclusions section with future research needs.

Edits

Line 322: Sentence too long and grammar needs improving.

Response: Done. The sentence is split into two sentence and rephrased for clarity.

Line 703: grammar needs improving.

Response: Done